

# A history-matching analysis of the Antarctic Ice Sheet since the last interglacial – Part 2: Glacial isostatic adjustment

Benoit S. Lecavalier[1], Lev Tarasov[1]

[1]Department of Physics and Physical Oceanography, Memorial University of Newfoundland, St. John's, Canada

Correspondence to: Benoit S. Lecavalier (b.lecavalier@mun.ca)

## Abstract

We present a glacial isostatic adjustment (GIA) analysis for a joint ice and GIA history matching of the Antarctic Ice Sheet (AIS) since the last interglacial. This was achieved using the
Glacial Systems Model (GSM) – which includes a glaciological ice sheet model asynchronously coupled to a viscoelastic earth model. A large ensemble of 9,293 simulations was conducted using the GSM. The history matching was against the AntICE2 database, which includes observations of past relative sea level, present-day (PD) vertical land motion, past ice extent, past ice thickness, borehole temperature profiles, PD geometry and surface velocity (Lecavalier et al., 2023). The 38
ensemble parameters of the GSM were history matched using Markov Chain Monte Carlo sampling that in turn employed Bayesian Artificial Neural Network emulators. The implications on the evolution of the AIS are detailed in a companion paper which predominantly focuses on the ice sheet component (Lecavalier et al. 2024). The history-matching analysis identified simulations from the full ensemble that are Not-Ruled-Out-Yet (NROY) by the data. This yielded a NROY
sub-ensemble of simulations consisting of 82-members that approximately bound past and present GIA and sea-level change given uncertainties across the entire glacial system. The NROY Antarctic ice sheet and GIA results represent the Antarctic component of the "GLAC3" global ice sheet chronology which acts as a primary input to GIA models of sea-level change.

Data-model comparisons are shown against a subset of the AntICE2 database which directly
constrains relative sea-level (RSL) change and GIA. A large variety of ice loading histories and Earth rheologies are evaluated against the available data. Significant spatial variability in Antarctic RSL and GIA are presented. The uncertainties affiliated with these inferences are large given the limited number of observational constraints which results in inferred RSL bounds with max/min ranges up to 150 m during the Holocene. Finally, estimates of PD rates of bedrock displacement
with tolerance intervals are presented and compared against reference Antarctic GIA studies. These previous Antarctic GIA studies are key inputs for geodetic studies of the contemporary AIS mass balance. We demonstrate that by adequately exploring glacial and rheological uncertainties against a comprehensive database, past studies have underestimated Antarctic GIA uncertainties across vast regions, while other sectors are now more narrowly constrained. This history matching
presents meaningful Antarctic GIA bounds of the rate of PD bedrock displacement with direct implications on mass balance estimates of the PD AIS.



# 1. Introduction

Large sectors of the Antarctic Ice Sheet (AIS) are undergoing accelerated mass loss (Seroussi et al., 2020; Masson-Delmotte et al., 2021). Of particular concern is that positive feedbacks can destabilize sectors of marine-based ice sheet, which raises concerns about the future evolution of the AIS (Pattyn and Morlighem, 2020; McKay et al., 2022). Even though the atmosphere and ocean directly influence AIS evolution, processes at the ice-bed interface can also dramatically impact ice dynamics. This is dictated by the basal environment which is characterized by several boundary conditions from basal topography, geothermal heat flux, and sediment distribution (Whitehouse et al., 2019). Moreover, the ice-bed interface is dynamic on a wide range of time-scales due to tectonic and volcanic activity, erosion and sedimentation, and glacial isostatic adjustment (GIA). GIA represents one of the key interactions between ice sheets and the solid Earth, which includes how the gravitational field and solid Earth respond to changes in ice and water load distribution. The GIA signal encompasses the continuous response of the solid Earth, gravity field, and relative sea-level to present and past ice sheet changes. Therefore, a robust understanding of GIA has implications on our understanding of AIS changes.

The GIA component included in ice sheet models vary significantly in terms of complexity (de Boer et al., 2017; Whitehouse, 2018). Glaciological AIS simulations have historically relied on a: simplified elastic lithosphere relaxed asthenosphere GIA models (e.g. Huybrechts, 2002; Whitehouse et al., 2012a; DeConto and Pollard, 2016; Pattyn, 2017); 2D GIA models based on a self-gravitating viscoelastic solid-Earth model with depth varying spherically symmetric earth rheology (Gomez et al., 2012; Briggs et al, 2013; Han et al., 2022) and 3D GIA models that account for lateral Earth structure (A et al., 2012; van derWal et al., 2015; Nield et al., 2018; Powell et al., 2021; Blank et al., 2021; Van Calcar et al., 2023).

To evaluate Antarctic GIA, one approach is to prescribe a predefined ice load history, although this neglects solid-ice sheet feedbacks on ice dynamics. A GIA model can be applied in series with ice sheet model output to produce higher fidelity GIA estimates than those computed exclusively within the ice sheet model (Whitehouse et al., 2012b; Lecavalier et al., 2014). Finally, self-consistent Antarctic GIA predictions are based on a fully coupled GIA component with an ice sheet model (Gomez et al., 2013; Briggs et al, 2014; Konrad et al., 2015; Pollard et al., 2017; Gomez et al., 2018; Han et al., 2022; Van Calcar et al., 2023). Although the fully coupled approach is effective at evaluating ice sheet and solid Earth processes and their feedbacks, the computational cost associated with these simulations typically prevents an adequate exploration of uncertainties across the glacial system. This remains a challenge when dealing with coupled ice sheet and 3D Earth models since the computational resources required prohibits a large-ensemble data-constrained analysis to infer the actual Antarctic GIA history rather than simply studying model behaviour from a small sample of simulations. Moreover, past studies generally relied on the use of a few observational constraints to evaluate model performance (Gomez et al., 2013, 2018; Pollard et al., 2017; Konrad et al., 2015) and/or conducted a limited exploration of system-wide parametric uncertainties (Ivins and James, 2005; Whitehouse et al., 2012b; Peltier et al., 2015). This directly limits the precise degree by which GIA or climate feedbacks might have actually impacted ice sheet instabilities in the actual past. Therefore, it is important to evaluate models against an observational constraint database that jointly consider coupled feedbacks between ice sheets and GIA.

The interaction between the solid Earth and the AIS through GIA is dictated by the rate and magnitude of ice sheet changes, and Earth's rheological properties. However, beneath the Antarctic continent there are large variations in rheological properties of the mantle (Ritzwoller et



al., 2001; Schaeffer and Lebedev, 2013; Heeszel et al., 2016; Shen et al., 2018; Lloyd et al., 2020).
These rheological variations define the viscosity of the mantle and the subsequent relaxation
timescales from a surface loading or unloading event. There are regions of apparently anomalously
low upper mantle viscosities in the Antarctic Peninsula and Amundsen Sectors that experience a
more rapid GIA response to mass loss relative to other Antarctic sectors (Nield et al., 2014; Barletta
et al., 2018). This implies that the present-day (PD) viscous GIA signal in regions with low
effective mantle viscosity is possibly more dominated by ice sheet changes over the last several
millennia rather than early deglacial ice sheet mass loss. As such, any approach to past inference
of AIS evolution should account for this lateral variation in effective earth viscosity.

The contemporary mass balance of the Antarctic ice sheet is inferred using a variety of geodetic
methods. Ice sheet changes can be monitored using satellite altimetry, radar imagery, optical
imagery, and gravimetry (e.g. Mouginot et al., 2017; Gardner et al., 2018; Smith et al., 2020;
Tapley et al., 2019; Velicogna et al., 2020; Sasgen et al., 2020). However, to infer the mass balance
of the AIS using these methods, GIA estimates with meaningful uncertainty estimates are required
(Shepherd et al., 2018; Otosaka et al., 2023). The Ice Sheet Mass Balance Inter-comparison
Exercise (IMBIE) initially reconciled these various satellite methods (Shepherd et al., 2012, 2018)
and has continued to provide mass balance estimates extending to 2020 (Otosaka et al., 2023). By
combining these different mass balance inference methodologies, confidence in both
contemporary mass balance and contributions to sea-level rise are enhanced. Uncertainties in
Antarctic GIA dominate the contemporary mass balance confidence intervals. As of now, all these
methods rely on Antarctic GIA inferences that have been derived with limited attention to full
system and observational uncertainties (Ivins and James, 2005; Whitehouse et al., 2012b; Peltier
et al., 2015). This  limitation will propagate to the inferred magnitude of PD mass balance. These
past reference GIA studies did not adequately consider system-model uncertainties when
quantifying data-model scores. At best, a limited exploration of parametric uncertainties on a very
narrow set of ice sheet and GIA model ensemble parameters was performed and no effort was done
to quantify the considerable impact of model structural uncertainties. A data-constrained AIS and
GIA model which accounts for complete uncertainties in the glacial system and Earth rheology is
necessary to provide accurate bounds on contemporary mass balance of the AIS.

## 2. Model description

The GSM consists of comprehensive ice dynamic, climate forcing, and glacial isostatic
components which are described in Tarasov et al. (submitted to GMD) and Lecavalier and Tarasov
(2024). To summarize the GSM includes: Hybrid SIA-SSA ice physics; subgrid grounding line ice
flux parameterization; dual power basal drag for hard bed and till sliding; ice shelf hydrofracturing
and ice cliff failure; ocean temperature dependent sub ice shelf melt parameterization; subgrid ice
shelf pinning point scheme; expanded climate forcing scenarios. An illustration showing the key
components of the GSM is found in Figure S1 of Lecavalier and Tarasov (2024).

The GSM is coupled to a glacial isostatic adjustment model of sea-level change based on a
self-gravitating viscoelastic solid-Earth model which calculates GIA due to the redistribution of
surface ice and ocean loads (Tarasov and Peltier, 1997). The Earth model rheology has a PREM
density structure (Dziewonski and Anderson, 1981) and an ensemble parameter controlled three
shell viscosity structure defined by the depth of the lithosphere, upper and lower mantle viscosity.
The GIA component shares many similarities to that used in Whitehouse et al. (2012b) for post-
processing modelled ice sheet chronologies, however, our GIA component is asynchronously
coupled to the ice sheet component. Considering GIA operates on longer timescales, the GIA





calculations are computed every 100 simulation years. To minimize the considerable computational cost of solving for a complete gravitationally self-consistent solution coupled with an ice sheet model (Gomez et al., 2010, 2013), a zeroth order geoidal approximation is used to account for the gravitational deflection of the sea surface. However, upon completing the full transient simulation, a gravitationally self-consistent solution is computed. The complete solutions are those that are compared against the GPS and RSL observations in Section 4. The continental scale transient Antarctic simulations over 205 ka have a 40 by 40 km horizontal resolution with the full sea-level solution having a spherical harmonic degree and order of 512.

As detailed in the accompanying study (Lecavalier and Tarasov, 2024), the Antarctic configuration of the GSM consists of 38 ensemble parameters. This represents the most comprehensive exploration of parametric uncertainties across the entire Antarctic glacial system of any study to date. A given simulation is defined by a parameter vector which consists of chosen values for each ensemble parameter. The ensemble parameters define the uncertainties in the climate forcing, mass balance, ice dynamics, and solid earth rheology. A total of three ensemble parameters defines the uncertainties in the viscosity profile of the solid Earth which directly impact GIA. Specifically, the lithospheric thickness, upper mantle viscosity, and lower mantle viscosity can respectively vary between 46 to 146 km, $0.1 \cdot 10^{21}$ to $5 \cdot 10^{21}$ Pa·s, and $1 \cdot 10^{21}$ to $90 \cdot 10^{21}$ Pa·s. GIA models simulate the response of the solid Earth due to present and past changes in surface loading from the redistribution of ice, water, and mantle material. The two primary inputs to a GIA model are a global ice chronology and the Earth rheology. The GIA model products will henceforth be referred to as GIA inferences which include past and PD bedrock deformation, geoid and RSL estimates. In this study, the GSM simulates AIS changes over the last 2 glacial cycles to minimize initialization uncertainties propagating into the last interglacial start of our history-matching interval. The GSM relies on several eustatic global sea-level forcing time series (e.g. Lambeck et al., 2014; Lisiecki and Raymo, 2005) when performing joint ice sheet and GIA calculations.

## 3. Methodology

As part of the history-matching analysis conducted in this study, simulations are ruled out given their inconsistency with observational constraints. This involves a data-model comparison that accounts for both data-system and system-model uncertainties to evaluate the performance of a simulation (Tarasov and Goldstein, 2021). The observational constraint database applied in this research is the Antarctic ICe sheet Evolution observational constraint database version 2 (AntICE2 database; Lecavalier et al., 2023). The AntICE2 database is to date the largest quality-curated database of Antarctic paleo-data based on a variety of data types that can be leveraged to constrain various facets of the Antarctic glacial system. The AntICE2 database (Fig. 1) consists of observations of past RSL, ice sheet thickness, ice sheet extent and PD observations of land motion from GPS measurements, ice core borehole temperature profiles, ice sheet geometry (Bedmachine version 2 Morlighem et al., 2020) and surface velocity (Mouginot et al., 2019). The GSM simulations are scored against the highest quality data in the AntICE2 database, with a predominant focus on tier-1 and 2 data. Tier-1 data has the greatest power to constrain the ice sheet and GIA model and is deemed the highest quality data. Generally, tier-2 data provides more granular detail on past changes and supplements tier-1 data. Tier-3 data correlates highly with the higher quality tier-1/2 data, therefore, it is excluded from the history-matching analysis and only used for visual comparison. In this study, the data-model comparison focuses on the RSL and GPS data given its relevance to GIA processes. All other data-model comparisons and discussions can



be found in the accompanying study (Lecavalier and Tarasov, 2024) which specifically focuses on the glaciological evolution of the AIS.

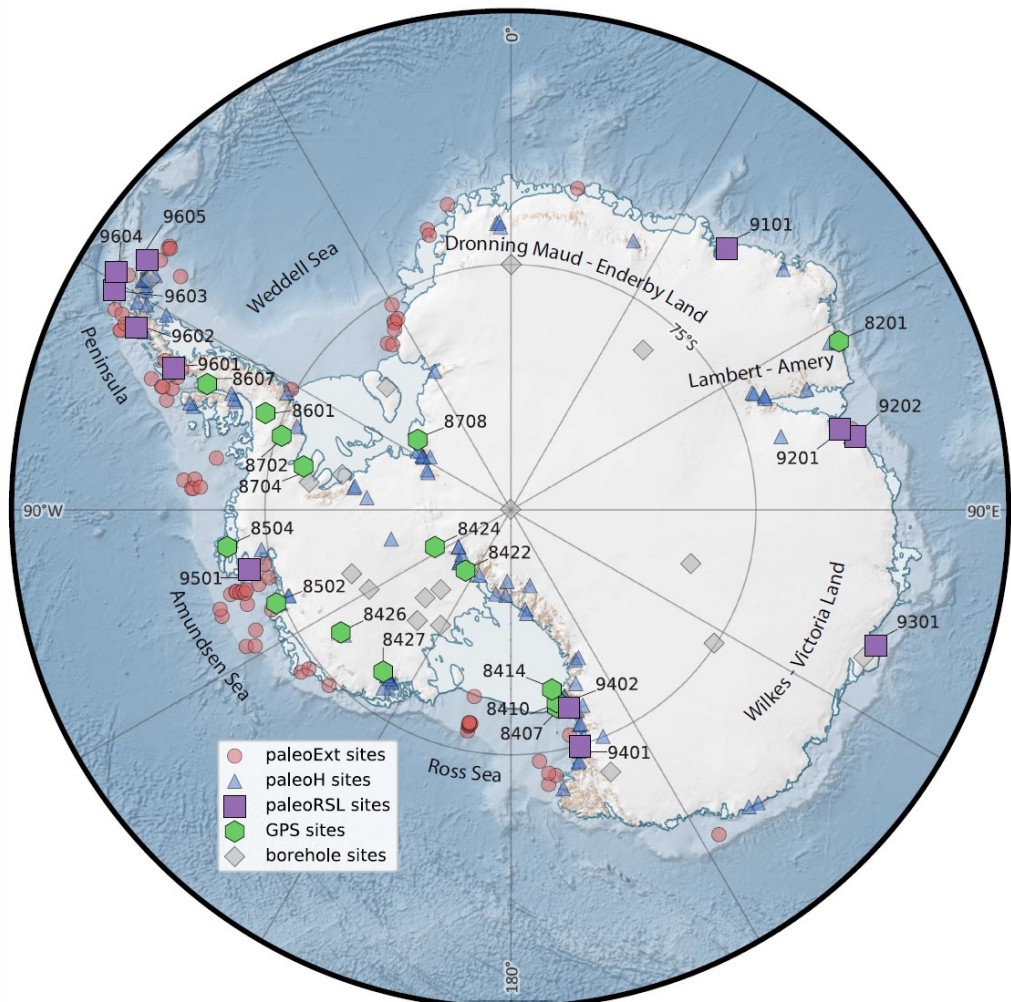

Figure 1: Antarctic continent and sector names mentioned in the study are shown alongside the Antarctic ICe sheet Evolution database version 2 (AntICE2) database (symbols). The data ID numbers for the paleoRSL and GPS data are shown. The remaining data ID information can be found in Figure 2 of Lecavalier et al. (2023). The Antarctic basemap was generated using Quantarctica (Matsuoka et al., 2021).

The paleoRSL and GPS data are inhomogeneously distributed across Antarctic in both space and time. The majority of the paleoRSL data spans the mid to late Holocene ages. The PD GPS measurements of bedrock displacement integrate the signal from several processes that operate on various time scales. The integrated time scale of the viscous relaxation due to ice and ocean unloading or loading depends on the viscosity of mantle material underlying the GPS station.



Therefore, the paleoRSL and GPS bedrock displacement data most meaningfully constrain the Holocene Antarctic GIA. There are few places in Antarctica that are deglaciated and preserved RSL proxy data. Similarly, there are a limited number of GPS observations which have a significant signal-to-noise ratio with a minimal elastic correction due to contemporary mass loss.

In this study we only provide a cursory overview of history matching, for a comprehensive description we point the reader to Tarasov and Goldstein, 2021 (a future submission will explicitly detail the exact history-matching methodology). This analysis yields a sub-ensemble of model simulations that are not-ruled-out-yet (NROY) by the data. The NROY sub-ensemble provides initial confidence intervals, minimum and maximum bounds on the probable evolution of the AIS

and GIA since the last interglacial.

As part of this study, several large-ensemble data-constrained analyses were iteratively performed to evaluate the model's ability to bracket the AntICE2 observational constraint database (Lecavalier and Tarasov, 2024). GSM simulations were applied to supervised machine learning of Bayesian Artificial Neural Networks (BANNs) to create an emulator of the GSM to efficiently

explore the parameter space. The history-matching methodology is diagrammatically illustrated in Figure S4 in Lecavalier and Tarasov (2024). The ensemble parameter prior ranges are based on experimentation, previous studies, expert judgement, and are initially kept wide as to not pre-emptively neglect any potentially relevant regions of the parameter space. When comparing a simulation to data, we carefully characterize the error model which combines all the errors

attributed to data-system (measurement and indicative meaning uncertainties) and system-model (structural) uncertainties to produce a meaningful implausibility score (Tarasov and Goldstein, 2021; Lecavalier and Tarasov, 2024; Tarasov et al., in prep). This includes a 5% structural bias error for RSL and a 1 mm/yr to 0.5 mm/yr bias error for PD vertical uplift due to the use of a global 2D earth rheology instead of a 3D Earth rheology based on the discrepancies between

corresponding results for regional 2D and 3D Earth modelling of Fennoscandia (Whitehouse et al., 2006; Whitehouse, 2009). However, a similar analysis for Antarctica is lacking.

As the robustness of history matching is predicated on complete sampling of the model parameter space, we use Markov Chain Monte Carlo (MCMC) to sample for the parameter vectors that are most likely to be consistent with the constraint data. To computationally enable the

required multi-million-point MCMC sampling, GSM ensemble output is used for supervised training and validation of BANNs to establish computationally efficient emulators of the GSM. Many BANNs were trained to predict specific targets (e.g. grounded ice volume and area, past ice extent, ice thickness, RSL, GPS uplift rates) given parameter vectors as input and site coordinates. The BANN architectures that proved to be effective for these targets are detailed in Tarasov et al.

(in prep). The BANN targets are key model metrics and/or specific predictions intended for data-model comparison. A sub-sample of parameter vectors from an order 100 set of MCMC converged sampling chains is in turn used to create an ensemble of GSM simulations. This combined MCMC and GSM ensemble iteration constitutes a "wave" of simulations that get added to the full ensemble.

The history-matching criteria rules out simulations with respect to the observations. As detailed in Tarasov and Goldstein (2021), a 3σ of total uncertainty threshold is the minimum generally used in history matching (and can go as high as 5σ). This is achieved by comparing key metrics of interest and model output against observational constraints and ruling out simulations which are inconsistent with the constraint data. Our implausibility threshold for inconsistency is a simulation-

data misfit score component value of between 3-σ and 4-σ of the total uncertainty (internal discrepancy, external discrepancy, and data uncertainty; see Table S1 in Lecavalier and Tarasov,



2024). In the case of the Antarctic GSM configuration, the primary metrics of interest were chosen to be: PD ice thickness root-mean-square-error for West Antarctic Ice Sheet (WAIS; which includes the Antarctic Peninsula Ice Sheet for simplicity), East Antarctic Ice Sheet (EAIS), and
floating ice; PD ice shelf area score; PD grounding-line position score along 5 transects; ice core borehole temperature profile score; GPS uplift rate score; past ice thickness score; past ice extent score; and past relative sea level score. To ensure an adequately sized NROY sub-ensemble, 3σ of the total uncertainty threshold was applied on all data type scores except for the following: past ice extent (3.5σ), floating ice RMSE (3.5σ), and relative sea-level scores (4σ). This gives an NROY
set of 82 simulations (and corresponding parameter vectors). This larger allowance with these three scores was justified given the model struggles to bracket a few observations in these data types, which resulted in ruling out nearly all simulations if imposing a 3σ threshold across all data types.

Previous ensembles of simulations were evaluated against the AntICE2 database and PD observations to verify that the observations are adequately bracketed by the GSM given
uncertainties. This led to an increase in the number of ensemble parameters, process additions to the GSM, and revisions to certain boundary conditions. Leading up to the final waves of ensembles, over 30,000 model simulations were performed as part of previous experimentation, sensitivity analyses, Latin Hypercube, and beta fit sampling of ensemble parameters. In the results section, we present the latest iterations of large-ensemble results based on the history-matching
analysis which consists of the final 9,293 simulations.

In addition to the history-matching analysis, an initial exploration on the potential impact of lateral Earth structure was conducted through a sensitivity analysis. Past studies have found that the spatially averaged upper mantle viscosity to be on the order of $10^{20}$ to $10^{21}$ Pa·s (Ivins and James, 2005; Whitehouse et al., 2012b) which is within the range evaluated as part of the history-
matching analysis. However, there are more recent estimates of an anomalously low upper mantle viscosity for the Antarctic Peninsula, Amundsen sector, and part of the Weddell and Ross Sea sector on the order of $10^{18}$ to $10^{19}$ Pa·s (Wolstencroft et al., 2015; Zhao et al., 2017; Barletta et al., 2018; Nield et al., 2018; Whitehouse et al., 2019). A high variance 17 member subset (HVSS) of simulations were selected from the NROY sub-ensemble according to key metrics of interest, such
as the AIS grounded ice volume during the Last Glacial Maximum (LGM) (Figure S7 in Lecavalier and Tarasov, 2024). This also included selecting simulations with minimum scores to certain data types. The ice load chronologies from the 17 members of the NROY sub-ensemble HVSS were subject to repeated GIA post-processing over a range of Earth models. Specifically, the lithospheric thickness and upper mantle viscosity was progressively decreased to 46 km and $5 \cdot 10^{18}$
Pa·s, respectively, to evaluate the impact of an anomalously low upper mantle viscosity on isostasy (Fig. S1 and S2). This experimental design isolates the Earth model sensitivity  at the cost of lost dynamical self-consistency between the ice history and earth model.



Figure 2: Paleo relative sea level data-model comparison where the grey shading are the full ensemble statistics. The solid and dashed black lines are the mean and min/max ranges for the not-ruled-out-yet (NROY) best fitting sub-ensemble. Simulations consisting of a high variance subset (HVSS) of the NROY sub-ensemble are shown in red. The 2σ and 1σ ranges are the nominal 95% and 68% ensemble intervals based on the equivalent Gaussian quantiles, respectively.





# 4. Results and Discussion

Below we present the full ensemble and NROY sub-ensemble against the AntICE2 observational constraints of most relevance to GIA: past RSL and elastic-corrected GPS measured rates of vertical land motion. Data-model comparison to the other constraints in AntICE2 are shown in Lecavalier and Tarasov (2024). The 2σ and 1σ ensemble ranges shown across several figures (e.g. Fig. 2-7) are the nominal 95% and 68% ensemble intervals based on the equivalent Gaussian quantiles (2.275 - 97.725%, Gaussian 2σ quantiles and 15.866 - 84.134% Gaussian 1σ quantiles).

There are a limited number of sites that record past RSL history across Antarctica and few sites with GPS measurements which are minimally contaminated by the elastic signal. There are considerably more observations that constrain the ice sheet evolution, thereby constraining the ice load history which acts as a primary input to GIA modelling. The other primary input is the Earth rheology which remains challenging to constrain given the limited observational constraints across a large spatial scale with inherent variation in lateral Earth structure. Data-model discrepancies are then due to some combination of the following: ice load history or Earth rheology, system-model uncertainties (e.g. lateral Earth structure), and data-system uncertainties (e.g. incorrect proxy indicative interpretation, underestimated elastic correction).

The Earth model ensemble parameters for the full ensemble and NROY sub-ensemble are shown in Figure S7 in Lecavalier & Tarasov, 2024. Relative to the full ensemble, the NROY sub-ensemble includes proportionately more simulations with upper mantle viscosities in the lower end range of $1 \cdot 10^{21}$ Pa·s to $5 \cdot 10^{21}$ Pa·s and lower mantle viscosities in the upper range of $20 \cdot 10^{21}$ to $70 \cdot 10^{21}$ Pa·s. This results in Earth model parameters that exhibit greater isostatic sensitivity on shorter time-scales, including those from smaller unloading events. Given the AntICE2 data that directly constrains rheological properties are primarily located in the WAIS and Antarctic Peninsula, this implies that the NROY sub-ensemble Earth model parameters are best suited for those sectors.

## 4.1 Data-model comparisons

There are a total of 12 sites across Antarctica that record past RSL change based on the quality curated data (Tier-1/2 quality data; Lecavalier et al., 2023). Figure 2 shows the full ensemble and NROY sub-ensemble against tier-1 and tier-2 paleo RSL data. There is a direct trade-off between improving the fit to one data type versus another in the database. This results in simulations which perform very well against just a single data type (e.g. paleo RSL data) to be ruled out if it performs poorly against any other (>3σ threshold).

The full ensemble brackets the RSL site in Dronning Maud-Enderby Land (site 9101) within observational uncertainty. The NROY sub-ensemble is also able to bracket the majority of the RSL observations at the Syowa Coast except for the two limiting dates between 8 and 9 ka, which constrain the sea-level highstand in the region. The amplitude of the NROY sub-ensemble mean RSL at Syowa Coast is quite low as compared to the data. Only simulations in the NROY sub-ensemble with the highest amplitude fit the data. The NROY RSL simulations demonstrate a high level of correlation between simulations, albeit with a different amplitude. A change in Earth structure in the region does considerably impact the amplitude of RSL change and time of decay (Fig. S1). Therefore, lateral Earth structure that corresponds to a lower upper mantle viscosity can produce a RSL fall consistent with the data. Although, there is limited evidence of lateral structure in this region. This suggests the unloading history and magnitude of ice loss could be responsible for the discrepancy with the sea-level highstand data in this area. The exposure age data





constraining the paleo ice thickness (paleoH) history in the region (i.e. site 1105:7 in Lecavalier and Tarasov, 2024) are all dated to the early to mid Holocene (9 to 6 ka). The NROY sub-ensemble bracket 6 of the 7 paleoH observations in the region. This highlights a potential issue in the climate forcing leading into the Holocene. For example, the ocean forcing drastically impacts the regional timing of unloading which leads to a RSL highstand of 15 meters above sea level (masl) at 7 ka instead of 20 masl at 8.5 ka. This combined with poorly resolve subgrid features (subgrid till basal trough or pinning points) off the coast of Syowa could impact the timing and magnitude of RSL highstand in the region.

In the Lambert-Amery sector, there are two RSL sites east of the Amery ice shelf that are both bracketed by the full ensemble and NROY sub-ensemble which demonstrate a RSL highstand of 7 to 9 masl around 7 ka (site 9201 and 9202; Fig. 2). With only the oldest lower limiting date in Larsemann Hills (site 9201) being inconsistent with the NROY predictions. The area surrounding these two RSL sites is also constrained by a proximal to grounding line constraint with an age of 10.5 ka (site 2201) which are bracketed by the NROY sub-ensemble. Among the NROY RSL predictions, there are some simulations that demonstrate RSL oscillations, with one highstand peak at 9 ka and another at 7 ka, which mirror the RSL data at Larsemann Hills except at the wrong amplitude. Simulations that produce such RSL oscillations in the region using an alternate Earth rheology can dampen the RSL amplitude in line with observations based on the Earth model sensitivity analysis (Fig. S1).

The Wilkes-Victoria Land represents the region with the fewest observational constraints along the margin of the PD ice sheet. At Windmill Island (site 9301), a limiting sea-level date and index point suggests a sea-level highstand of ~30 meters at 8 ka which is not bracketed by either the full ensemble or NROY sub-ensemble. The NROY sub-ensemble produces a sea-level highstand at 7 ka between 7 to 16 masl which is half of the amplitude necessary to be consistent with the few observations in the region. Given the full ensemble does not achieve the necessary amplitude needed to capture the observations at Windmill Island regardless of the range of Earth rheology considered in the history-matching analysis and Earth model sensitivity analysis, the discrepancy can be attributed to the ice load chronology. The region is poorly constrained with only the Law Dome borehole temperature profile which provides a minimal constraint on isostasy and ice unloading across the continental shelf. A broadening of viable ice reconstructions in the Windmill Island region is necessary to produce a larger ice unloading event over the deglaciation to match the observed sea-level amplitude. This suggests the climate forcing or basal conditions may lack adequate regional degrees of freedom to produce a sufficiently larger ice load in the region, and subsequent deglaciation timing to reach the observed sea-level highstand in the paleo RSL data.

Along the Transantarctic Mountains, there are two RSL records (site 9401 and 9402) which are bracketed by the full ensemble of RSL simulations. The NROY sub-ensemble brackets the sites except for the highest sea-level observations at Terra Nova Bay (9401). The region is topographically complex with subgrid valley glaciers that are poorly resolved in the GSM. This is a recurring challenge performing a data-model comparison to paleoH data in the region which can manifest in an inaccurate ice unloading history. Moreover, this region of the Transantarctic Mountains has an anomalous low viscosity zone in the upper mantle which has consequences on the viscous response to past load changes (Whitehouse et al., 2019). The NROY HVSS Earth model sensitivity analysis demonstrates that by lowering the upper mantle viscosity in this region, a more rapid viscous response to ice unloading can reach the peak observed RSL at Terra Nova Bay.



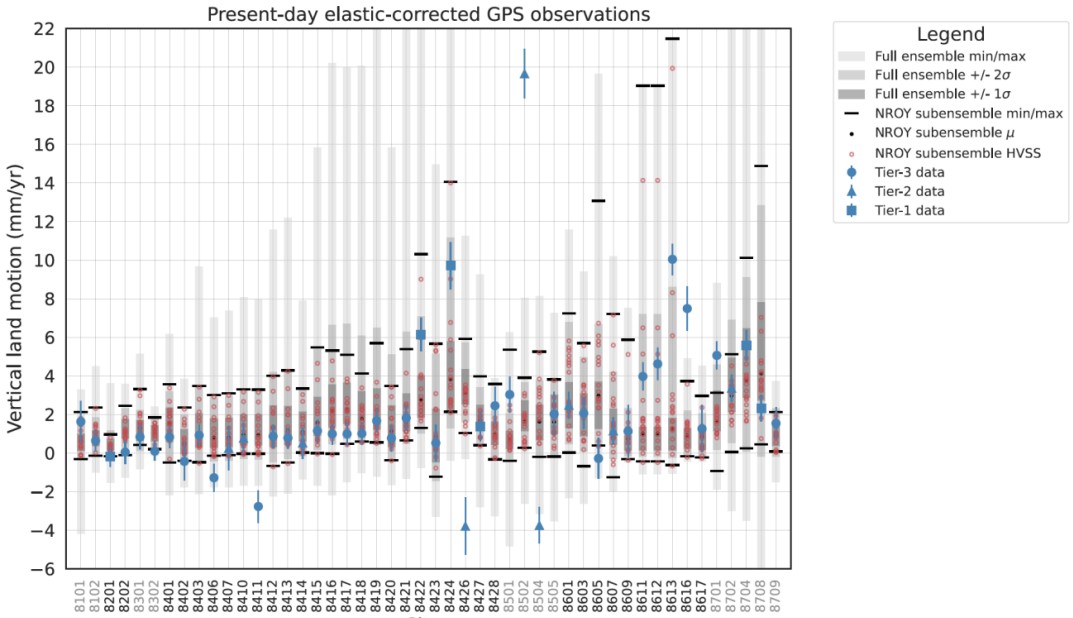

Figure 3: Global Positioning System elastic-corrected rate of bedrock displacement data-model comparison for tier-1/2/3 data in AntICE2. The grey shading represents the min/max, 1σ and 2σ ranges of the full ensemble. The solid black circles and lines are the mean and min/max ranges for the not-ruled-out-yet (NROY) sub-ensemble. Simulations consisting of a high variance subset (HVSS) of the NROY sub-ensemble are shown as red circles. The 2σ and 1σ ranges are the nominal 95% and 68% ensemble intervals based on the equivalent Gaussian quantiles, respectively.

In the Amundsen sector there is a RSL record which demonstrates rapid RSL fall of ~15 m over 4 ka (Fig. 2). The NROY sub-ensemble narrowly fails to bracket the sea-level observations by the upper bound RSL predictions at Pine Island Bay by ~1 m. The full ensemble does manage to bracket the RSL observations. As such, simulations that produce the appropriate amplitude of RSL fall at Pine Island Bay are ruled out when compared against the entirety of the AntICE2 database. The region has extensive paleoH/Ext observations, which points to issues with the Earth rheology in the region. Considering the region has lateral Earth structure with a lower viscosity with respect to the interior of the WAIS, the amplitude and rate of RSL fall could be considerably impacted since the GSM operates with a spherically symmetric Earth model. A single viscosity profile is applied across Antarctica which biases regions that are data rich. The data rich sections in the AntICE2 database, particular with regards to RSL and GPS data, are in West Antarctica, Antarctic Peninsula, and Transantarctic mountains, which are coincidentally regions that have anomalous lateral viscosity structures (Whitehouse et al., 2019). Therefore, evaluating a coupled ice sheet and 3D GIA Earth model could delineate the potential GIA feedbacks impacting ice dynamics and ice mass loss in Pine Island Bay. However, the HVSS Earth model sensitivity analysis does not improve the fits at Pine Island Bay which suggests limitations with the NROY loading histories.



The remaining RSL data is in the Antarctic Peninsula (site 9601 to 9605) document a RSL fall of ~20m from the mid Holocene to present. Generally, the full ensemble and NROY sub-ensemble struggles to bracket some notable observations. The NROY sub-ensemble struggles to produce a rapid late Holocene RSL fall as reported at Byers Peninsula (9603), King George Island (9604), and Joinville Island (9605). The model does not appear to produce the necessary magnitude of mass loss sufficiently late to result in the GIA required to capture these late Holocene observations, regardless of the chosen viscosity structure (Fig. 2 and S1). This is particularly the case at Byers Peninsula and King George Island where there is a misfit of over 10 m at 2 ka and 8 m at 7 ka, respectively. This suggests the climate forcing enabling thick ice in the region also fails to appropriately deglaciate the local region late enough. Additionally, the regional topography consists of a poorly resolved subgrid archipelago. The data at site 9602 is not shown considering it is a singular low quality tier-3 data point. It is a max limiting age at ~7 m with large temporal uncertainties ranging between 1.25 to 2.75 ka. This data suggests a similar, albeit poorly constrained rapid late Holocene RSL fall, like other RSL data in the region. However, the HVSS Earth model sensitivity analysis demonstrates that a dramatic late Holocene RSL fall exceeding 7 m can be produced using lower upper mantle viscosities (Fig. S1). This highlights that with a late Holocene unloading event, a rapid RSL fall of considerable amplitude is achievable in this region. Unfortunately, the necessary unloading events at the northern tip of the Antarctic Peninsula are not represented in the NROY sub-ensemble to the extent needed to address the outstanding discrepancies at site 9603, 9604, and 9605.

With respect to the elastic-corrected GPS observations of vertical land motion, the full ensemble and NROY sub-ensemble broadly bracket the observations (Fig. 3). Although there are a few exceptions, our focus will remain on the highest quality tier-1 and tier-2 data. The quality GPS data are predominantly located in the Ross Sea, Amundsen, Antarctic Peninsula, and Weddell Sea sectors, with a lone site in the Lambert-Amery sector. Tier-1 and tier-2 GPS observations that are not bracketed by the simulations tend to misfit both the full ensemble and NROY sub-ensemble at 3 distinct sites 8426, 8504, and 8502 in or near the Amundsen Sea sector. The Amundsen Sea sector has an anomalous low viscosity zone in the upper mantle which is not differentiated in the spherically symmetric GIA model. The HVSS Earth model sensitivity analysis does demonstrate that by considering a low upper mantle viscosity, elastic-corrected GPS predictions are captured at site 8406, 8411, 8504, 8616, and 8701 (Fig. S2) which are regions with inferred anomalously low viscosity structure. As the upper mantle viscosity is decreased by several orders of magnitude, this can significantly increase or decrease the amplitude of the GIA response depending on the temporal proximity of the unloading event. None of the HVSS Earth model sensitivity simulations capture the GPS bedrock trends (-4 mm/yr at site 8426, and 19 mm/yr at site 8502; Fig. S2), even though such amplitudes are attainable at other sites. Alternatively, the elastic corrections applied to the GPS data could be underestimated, particularly given their full uncertainties are ill-defined with its limited reliance on the input contemporary mass balance estimates (Martín-Español et al., 2016; Sasgen et al., 2017). Negative vertical land motion at 8426 suggests regional loading not represented by the elastic correction and/or GSM simulations. Conversely, 8502 with its exceedingly high elastic-corrected uplift rate indicates significant mass loss in the late Holocene. This suggests that the GSM might have insufficient degrees of freedom in the regional climate forcing and basal environment to produce a sufficiently late ice load scenario to reconcile these remaining discrepancies.





## 4.2 Glacial isostatic adjustment model predictions

The history-matching result being the NROY sub-ensemble is a product of ruling out simulations that were inconsistent (within 3 to 4σ) with all the data types in the AntICE2 database. A given data type in the database constrains the Antarctic GIA ensemble results to various degrees since some data types are direct constraints on GIA (e.g. bedrock displacement) while others are indirect (e.g. load history). The NROY sub-ensemble brackets the majority of the AntICE2 database with some limited outstanding exceptions discussed above and in Lecavalier and Tarasov (2024). The full ensemble is sieved on a data type basis which avoids the need for any inter data type weighing. History matching on its own does not produce a probability distribution of chronologies. Throughout this study, we present the NROY sub-ensemble nominal 2σ range since studies that apply GIA corrections are typically interested in accounting for nominal 2σ uncertainties. Moreover, by visualizing 2σ ranges one avoid any one outlier simulation from dominating the visualization. However, these nominal ranges should not be confused with traditional Gaussian confidence intervals and the reader is encouraged to also consider the complete NROY sub-ensemble min and max GIA and RSL ranges shown in Figure S3-S6.

The NROY sub-ensemble spatial RSL bounds during the deglaciation are shown in Figure 4. The figure demonstrates the spatial variability in Antarctic RSL during the deglaciation. The NROY sub-ensemble RSL ranges (centre and rightmost columns of Fig. 4) can be quite wide considering how data-poor the Antarctic continent is in comparison to other currently or previously glaciated regions. Based on the NROY sub-ensemble mean, the range in RSL change surrounding the AIS during the local LGM (~15 ka) is -90 m off the continental shelf to 20 m near the Ronne-Filchner ice shelf grounding line (Fig. 4a). Similarly, spatial RSL variability during the Holocene for the NROY sub-ensemble mean range between -25 to 63 m (Fig. 4d). When comparing the 2σ range surrounding the AIS, there are RSL differences of 200, 150, and 50 m at 15, 10, and 5 ka, respectively (Fig. 4). This emphasizes the large regional RSL uncertainties in the entire glacial system. The RSL mean and ranges of the NROY sub-ensemble reflects the regional AntICE2 data density which constrains the range of viable ice load histories and Earth rheologies. Some regions can only load so much ice during the LGM because of a limited continental shelf. This directly limits the maximum possible ice load in certain regions which impacts the subsequent deglacial GIA response.

The PD rate of vertical land and geoid displacement due to past changes in ice load demonstrate the background viscous GIA signal (Fig. 5 and Fig. 6). The range presented by the NROY sub-ensemble can be down to -3 mm/yr for wide regions in the interior of the EAIS or WAIS; conversely, they can be beyond 10 mm/yr across the WAIS. A few reference simulations (RefSim1 = nn61639, RefSim4 = nn63807, RefSim5 = nn64802) are shown in Figure 5 and Figure 6 to convey GIA estimates at PD based on individual simulations rather than solely presenting sub-ensemble statistics. The reference simulations are all in the NROY sub-ensemble, where RefSim1 has the minimum score across all the data types in AntICE2, RefSim4 has the minimum GPS score, and RefSim5 has the joint minimum score across all the paleo data types in AntICE2.





Figure 4: The not-ruled-out-yet (NROY) sub-ensemble mean (left), minus 2σ (middle), and plus 2σ (right) regional Antarctic RSL during the deglaciation at 15 ka (top), 10 ka (middle), and 5 ka (lower). The 2σ ranges are the nominal 95% ensemble intervals based on the equivalent Gaussian quantiles.

485

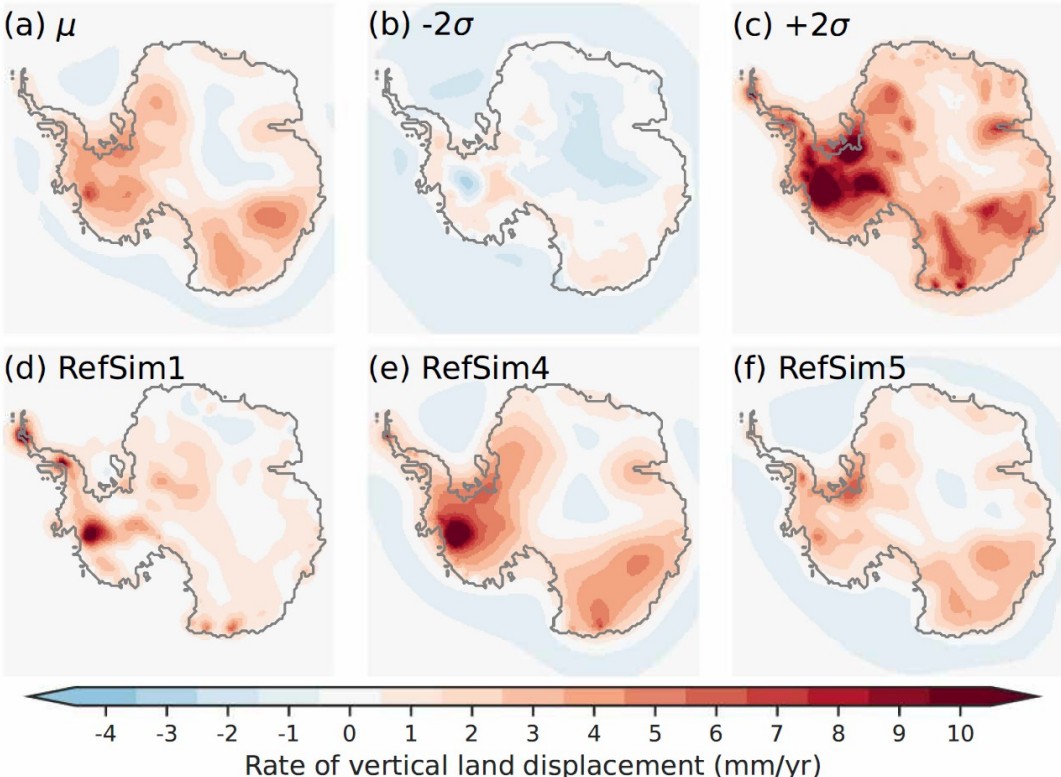

Figure 5: Present-day rate of bedrock displacement for the not-ruled-out-yet (NROY) sub-ensemble a) mean, b-c) minus and plus 2σ. Three glaciology self-consistent simulations chosen from a NROY high variance subset (HVSS): d) RefSim1 (NROY member with minimum score across all the data types in AntICE2); e) RefSim4 (NROY member with minimum GPS score); f) RefSim5 (NROY member with joint minimum score across all the paleo data types in AntICE2). The 2σ ranges are the nominal 95% ensemble intervals based on the equivalent Gaussian quantiles.

The NROY sub-ensemble demonstrates the wide range of viable PD GIA uplift rate estimates when one accounts for uncertainties across the entire glacial system and data-constrain a model against a comprehensive observational constraint database. There are three primary GIA reference models (IJ05_R2 - Ivins and James, 2005; W12a - Whitehouse et al., 2012b; ICE-6G_D - Peltier et al., 2015) applied across the IMBIE studies (Shepherd et al., 2018; Otosaka et al., 2023). These three reference GIA inferences have been used to produce a minimum and maximum range that represent nominal 2σ bounds on the PD GIA corrections when inferring contemporary AIS mass balance (Fig. 7). Comparison of these bounds with those from our history matching (Fig. 7) arguably provide an indication of where IMBIE GIA uncertainties are over and under estimated. The most prominent area where the reference GIA inferences have underestimated PD uplift rate uncertainties are in the Amundsen sector (Fig. 7d). This area suggests that uplift rates can exceed 10 mm/yr (Fig. 5c) which is not captured by the three reference GIA inferences (Fig. 7b). This corresponds to a negative geoid trend of -2.5 mm/yr across the Amundsen sector due to continued late Holocene mass loss (Fig. 6b). Moreover, significantly more observational constraints have





been collected in the Amundsen sector since the three reference GIA inferences were originally published. This includes past ice extent data along the Pine Island-Thwaites paleo-ice stream trough, and past ice thickness data near the Pine Island and Thwaites glacier (Fig. 1). Elastic-corrected GPS observations in the Amundsen sector indicate an uplift rate of ~20 mm/yr (site 8502). Finally, the latest RSL observations near Pine Island Bay propose a late and significant sea-level fall over the mid to late Holocene (site 9501 - Braddock et al., 2022) with implications on PD GIA estimates. These results demonstrate the current limitations of assessed IMBIE uncertainty ranges given its reliance three reference GIA inferences which were not designed to assess predictive uncertainties. Considering that at present, the Amundsen sector is undergoing by far the most mass loss across Antarctica (Shepherd et al., 2018; Otosaka et al., 2023), our revised PD GIA estimates and uncertainty range has significant consequences on the inference of the magnitude of mass loss across the West Antarctica and its corresponding confidence intervals.

The NROY sub-ensemble HVSS Earth model sensitivity analysis resulted in 153 simulations that were compared to the RSL and elastic corrected GPS observations to assess whether heterogeneity in Earth structure can potentially explain any outstanding discrepancies. This HVSS Earth model sensitivity analysis reveals that some of the previous RSL and GPS data not bracketed by the NROY sub-ensemble are captured if one considers an alternate regional Earth rheology (Fig. S1 and S2). The potentially rectification of some data-model discrepancies by an anomalously low upper mantle viscosity does not mean that the NROY sub-ensemble suddenly brackets this data. It simply demonstrates the plausibility that lateral Earth structure could reconcile some of these data-model discrepancies. Based on the Earth rheology sensitivity analysis, a more appropriate structural error is needed to study Antarctica GIA when using a 2D Earth model since the current specification is based on the discrepancies between 2D and 3D Earth models for Fennoscandia which appear to be significantly smaller than for Antarctica. To better address this question will require a more involved specification of the structural uncertainty attributed to lateral Earth structure or a history matching analysis with a coupled 3D GIA Earth model. Moreover, the upper mantle viscosity sensitivity analysis shows that regions with a low viscosity zone can exhibit significant sensitivity to recent loading or unloading. Thus, the NROY sub-ensemble GIA predictions likely underestimate the uplift and geoid rate bounds in specific regions with anomalous viscosity structure. Addressing these issues in future research is critical since GIA corrections and their uncertainties are used to infer contemporary AIS mass balance.

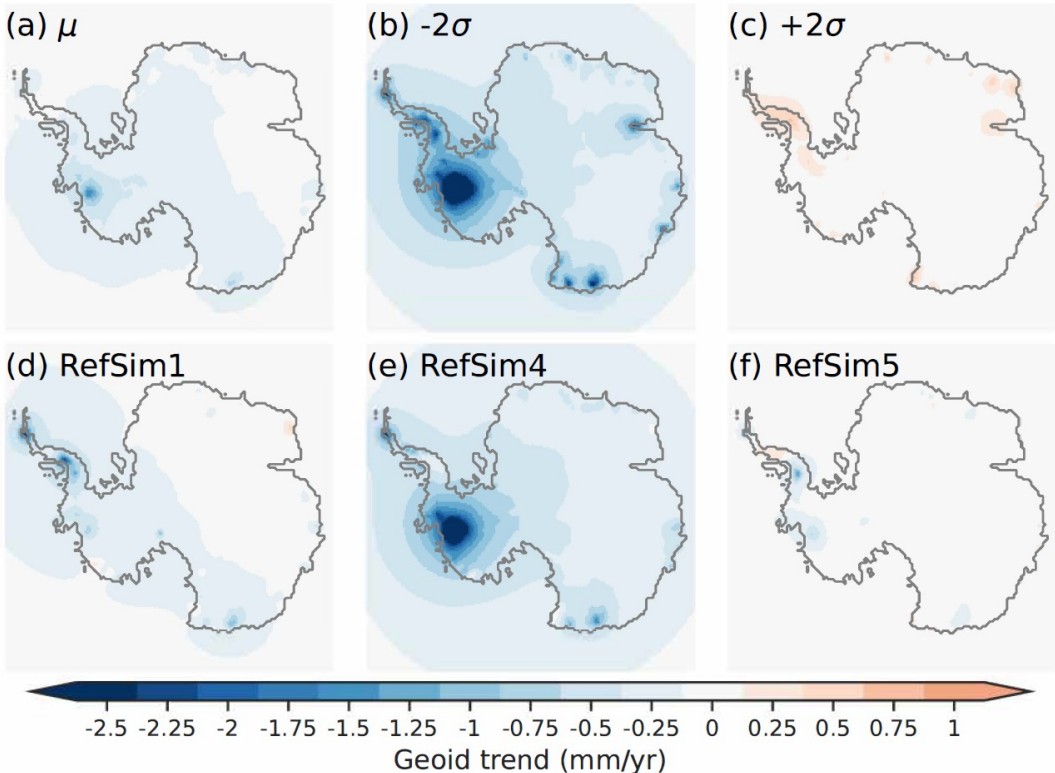

Figure 6: Present-day Antarctic rate of geoid displacement for the not-ruled-out-yet (NROY) sub-ensemble a) mean, b-c) minus and plus 2σ. The geoid trend presented above only includes the Antarctic contribution to geoid perturbations and does not include global eustatic contributions. Three glaciology self-consistent simulations chosen from a NROY high variance subset (HVSS): d) RefSim1 (NROY member with minimum score across all the data types in AntICE2); e) RefSim4 (NROY member with minimum GPS score); f) RefSim5 (NROY member with joint minimum score across all the paleo data types in AntICE2). The 2σ ranges are the nominal 95% ensemble intervals based on the equivalent Gaussian quantiles.

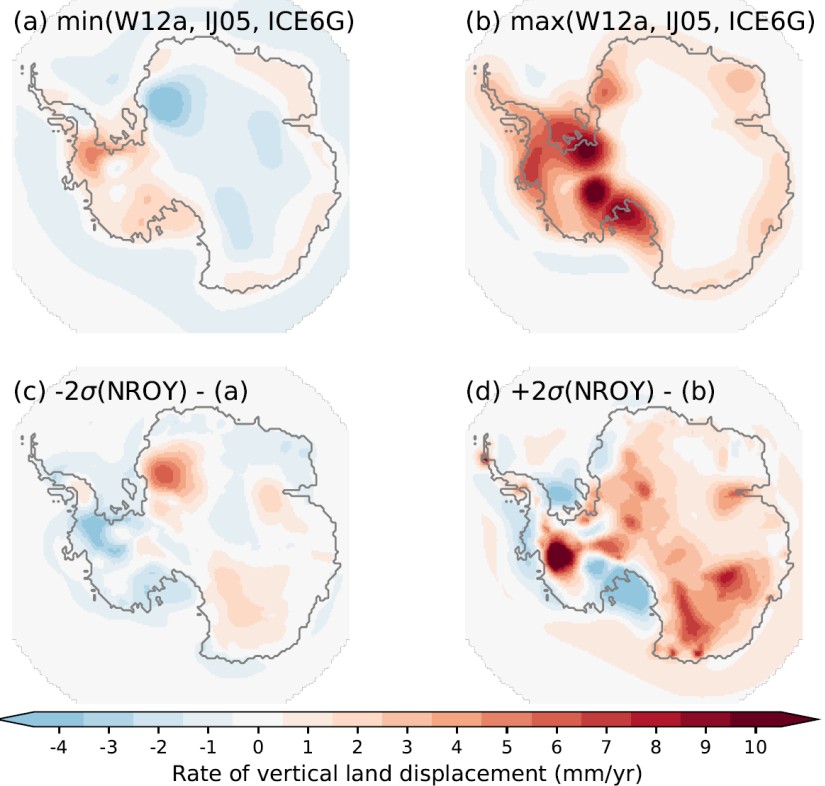

Figure 7: The (a) minimum and (b) maximum bounds for the PD rate of bedrock displacement for the three reference Antarctic GIA inferences (IJ05_R2 - Ivins and James, 2005; W12a - Whitehouse et al., 2012b; ICE-6G_D - Peltier et al., 2015). These three GIA inferences represent nominal 2σ bounds on the PD GIA corrections applied in the IMBIE studies to infer contemporary mass balance of the AIS. (Shepherd et al., 2018; Otosaka et al., 2023). The not-ruled-out-yet (NROY) sub-ensemble 2σ (c) lower and (d) upper bounds minus the respective bounds of the three reference GIA inferences. The differences shown in (c) and (d) demonstrate regions where the three reference GIA inferences underestimate PD GIA uncertainties or where the NROY sub-ensemble better constrain the regional GIA response relative to the three reference GIA inferences.

## 5. Conclusion

In this study a sub-ensemble of Antarctic GIA inferences is presented based on a history-matching analysis of the GSM against the AntICE2 database. The fully coupled glaciological and GIA model was used to generate a full ensemble consisting of 9,292 Antarctic simulations spanning the last 2 glacial cycles. BANNs were trained to emulate the GSM for rapid exploration of the parameter space via MCMC sampling. Simulation results were scored against past relative sea level, PD vertical land motion, past ice extent, past ice thickness, borehole temperature profiles, PD geometry and surface velocity. The scores were used in the history-matching analysis to rule out simulations that were inconsistent with the data given observational and structural uncertainties, thereby a NROY sub-ensemble (N=82) that bound past and present GIA and sea-



level change was generated. The NROY sub-ensemble of AIS results represent a collection of not-
ruled-out-yet Antarctic components for the global GLAC3 ice sheet chronology.

The NROY sub-ensemble AIS chronologies represent the Antarctic component in the GLAC
global ice sheet chronology. This research enables the upcoming evaluation of global RSL
predictions and the Antarctic far-field sea-level contributions during the last interglacial, LGM,
and deglacial melt water pulses. The AIS NROY sub-ensemble chronologies are constrained by
near-field observations. Evaluating the updated global ice sheet chronology against far-field RSL
observations would in turn constrain the AIS NROY sub-ensemble by said far-field data,
potentially ruling out additional AIS simulations that are currently in the NROY sub-ensemble.
This future work could leverage 3D Earth GIA models to formally evaluate lateral Earth structure
and its impact on far-field and near-field RSL predictions.

Given that our history matching accounts for data-system and system-model uncertainties to a
much deeper extent than any previous AIS study, the NROY sub-ensemble provides the most
credible bounds to date on actual Antarctic GIA. As such, our analysis demonstrates that previous
Antarctic GIA studies have underestimated the viscous deformation contribution to PD uplift rates
due to past ice sheet changes across several key regions. This is particularly the case in the
Amundsen sector, an area currently undergoing significant mass loss, which has a large range of
viable PD GIA estimates. Our NROY set of chronologies will therefore facilitate more accurate
inference of the PD mass balance of the AIS, including for vulnerable marine-based regions.

## Author Contribution

B.S.L. and L.T. led and designed the study. B.S.L. wrote the manuscript with editorial input from
co-author. B.S.L. and L.T. ran the model simulations, processed the dataset and simulation output.
B.S.L. and L.T. performed the data-model analysis. L.T. conducted the BANN training and
MCMC sampling. B.S.L. visualized the model results.

## Competing interests

The authors declare that they have no conflict of interest.

## Acknowledgements

Support provided by Canadian Foundation for Innovation, the National Science and Engineering
Research Council, and ACEnet. This research has been supported by an NSERC Discovery Grant
held by Lev Tarasov (number RGPIN-2018-06658) and is a contribution to the PalMod project
funded by the German Federal Ministry of Education and Research (BMBF).

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
