# Peer review of "A history-matching analysis of the Antarctic Ice Sheet since the last interglacial – Part 2: Glacial isostatic adjustment"

_EGUsphere, 2024_

## Author Comment (AC1)

We would like to thank the reviewer for their comments, suggestions, and feedback. This response aims to address any comments raised by the reviewer. Our responses are embedded below and are shown in orange.

**Response to referree comments #1**

The work of Lecavalier and Tarasov assesses uncertainties in the solid Earth's response to loading and unloading of ice (and consequently the oceans) in the Antarctic. The authors show that there is a much larger uncertainty and different spatial patterns than previously estimated by the more popular Glacial Isostatic Adjustment (GIA) models, which are used to estimate the present-day AIS mass balance. Their assessment is done by history-matching a large ensemble of simulations using the Glacial Systems Model (GSM), producing a Not-Ruled-Out-Yet (NROY) subset that is further used as input to *adhoc* simulations of GIA using a more faithful solid-Earth model. The GSM ensemble is the same presented in a companion paper. They further use the NROY ensemble results to discuss implications for the climate and GIA (including when that means a limitation of the used forcings or solid-Earth models and parameter ranges).

Overall, the paper structure is mostly clear and easy to follow with just some points where the text could be improved, sometimes by rewriting confusing paragraphs, sometimes by clarifying some technical parts. Below I make some general remarks with suggestions to improve the overall state of the paper before it can be published, provide technical/editorial suggestions line by line, and finally comment on how to improve some of the figures presented.

I hope the authors find my comments useful, and I look forward to seeing a revised and improved version of this manuscript.

**General remarks**

1. The introduction adequately provides the background necessary to contextualise the paper, but it lacks a proper ending pointing the reader to what research question(s) the study aims to address. Please add a final paragraph or a couple of sentences framing how the present study fits into the picture provided, and what its goals are.

   A couple sentences were added at the tail end of the introduction to better frame the research goals of the manuscript.

   "Part two of this study presented below aims to quantify the evolution of the AIS with a specific focus on GIA and sea-level change by properly exploring uncertainties in the glacial system. By applying a history-matching methodology, bounds are generated which define uncertainty ranges on AIS, GIA, and sea-level change that illustrates the limitation of simply focusing on a few best fitting models with direct consequences on how we understand the PD AIS. As part of the history-matching analysis, a sub-ensemble of AIS simulations are chosen to represent the Antarctic component and its respective uncertainties within the global ice sheet chronology GLAC3 for future analysis."

2. As per TC's guidelines, papers that are submitted or in prep (i.e., not yet available and without a DOI) cannot be cited. This needs to be rectified before the manuscript can be

published. I suggest acting on it now instead of waiting for the same issue to be pointed out by the typesetting or copy-editing teams.

The manuscripts that were cited as submitted or in preparation now have a doi and are in press. The GSM description paper was submitted on Sept 20th 2024, a pre-print is available with a permanent DOI at https://doi.org/10.5194/gmd-2024-175. Part 1 of this study is in press and a pre-print can be found at: https://doi.org/10.5194/tc-19-919-2025. The Tarasov & Goldstein (2021) study can be found at: https://doi.org/10.5194/cp-2021-145.

3. Considering I am no GIA expert, and that this paper is likely targeted at paleoglaciologists and ice sheet modellers as well, I personally feel that the GIA model description part is quite confusing and a bit unstructured, and could be improved. It would be beneficial to this manuscript if the authors invested some time in improving the flow of the model description section (especially the last two paragraphs), rearranging some of the sentences to make the sequence of information presented more logical (e.g., not going back and forth between the GIA and ice sheet components) and adding some clarifications to the more technical terms (e.g., PREM structure). I believe such changes would provide a much better context for the results, and aid the non-GIA experts who would likely be interested in this paper.

It is challenging to talk about GIA and ice sheet processes in isolation given they are a coupled system. The text was revised to consolidate the majority of the GIA description to a single paragraph (see below). This manuscript only provides an overview of the model description and provides the necessary information to understand the GIA component, resolution, initialization, and ensemble parameters. As stated in the paper, we recommend that the reader look at the complete model description paper for more detail:

Tarasov, L., Lecavalier, B. S., Hank, K., and Pollard, D.: The glacial systems model (GSM) Version 24G, Geosci. Model Dev. Discuss. [preprint], https://doi.org/10.5194/gmd-2024-175, in review, 2025.

    "GIA models simulate the response of the solid Earth due to present and past changes in surface loading from the redistribution of ice, water, and mantle material. The two primary inputs to a GIA model are a global ice chronology and the Earth rheology. The GIA model products will henceforth be referred to as GIA inferences which include past and PD bedrock deformation, geoid and RSL estimates. The GSM is coupled to a glacial isostatic adjustment model of sea-level change based on a self-gravitating viscoelastic solid-Earth model which calculates GIA due to the redistribution of surface ice and ocean loads (Tarasov and Peltier, 1997). The Earth model rheology has a standard Preliminary Reference Earth Model (PREM) density structure (Dziewonski and Anderson, 1981) and an ensemble parameter-controlled three-shell viscosity structure defined by the depth of the lithosphere, upper and lower mantle viscosity. A total of three ensemble parameters defines the uncertainties in the viscosity profile of the solid Earth which directly impact GIA. Specifically, the lithospheric thickness, upper mantle viscosity, and lower mantle viscosity can respectively vary between 46 to 146 km, $0.1 \cdot 10^{21}$ to $5 \cdot 10^{21}$ Pa·s, and $1 \cdot 10^{21}$ to $90 \cdot 10^{21}$ Pa·s. The GIA component shares many similarities to that used in Whitehouse et al. (2012b) for post-processing modelled ice sheet chronologies, however, our GIA component is asynchronously coupled to the ice sheet component. Considering GIA operates on longer timescales, the GIA calculations are computed every 100 simulation years. To minimize the considerable computational cost of

solving for a complete gravitationally self-consistent solution coupled with an ice sheet model (Gomez et al., 2010, 2013), a zeroth order geoidal approximation is used to account for the gravitational deflection of the sea surface. However, upon completing the simulation, a gravitationally self-consistent solution is computed using the AIS simulation as part of the global ice sheet chronology GLAC3. The complete solutions are those that are compared against the GPS and RSL observations in Section 4. The full continental scale transient Antarctic simulations over 205 ka have a 40 by 40 km horizontal resolution with the full sea-level solution having a spherical harmonic degree and order of 512."

4. The authors offload most of the explanation regarding scoring the simulations to two other papers: One that is "in prep", and another that is an exceedingly lengthy pre-print which was never accepted for publication. The "in prep" manuscript is provided as part of the review process, which is much appreciated (I actually found it very interesting and look forward to seeing it eventually published). Still, it is very much in preparation, and I could only get a general grasp of how the scoring was done. Considering that details regarding the scoring are not the focus of the manuscript under review, and "in prep" manuscripts cannot be cited, I would only ask that the authors explain slightly better why NROY simulations (or the entire ensemble, actually) do not bracket some observations, as evident in Figs 2 and 3. Is it because by choosing e.g., 3.5 or 4sigma means the "allowed variability" is actually larger than the ensemble variability itself? And what does the sigma refer to? Is it simply the standard deviation of the metric(s) being shown in the graphs?

There are a variety of approaches to data-model scoring (e.g. Briggs et al., 2013; Ely et al., 2019) and the one applied in this study is broadly described in Tarasov & Goldstein (2019) (https://doi.org/10.5194/cp-2021-145). Therefore, another ice sheet modeller could leverage their ice sheet model of choice to achieve a history-matching analysis. This manuscript is already exceeding lengthy, hence why we rely on citations and opted to exclude an exhaustive model description and data scoring methodology section. An accepted pre-print for part 1 which includes a description of the history-matching scoring and sigma thresholds can be found at: https://doi.org/10.5194/tc-19-919-2025. Furthermore, the complete details of the history matching implementation is a whole paper on its own and an asset was provided for additional context. The results and discussion section details the misfits shown in Figure 2 and 3. As per the methodology section: "Our implausibility threshold for inconsistency is a simulation-data misfit score component value of between 3-σ and 4-σ of the total uncertainty (internal discrepancy, external discrepancy, and data uncertainty; see Table S1 in Lecavalier and Tarasov, 2025)."

5. The GLAC3 chronology comes totally out of the blue, being mentioned only in the abstract and conclusions. All I can gather is that it stems from the NROY ensemble, but no other context is provided. It would be worth contextualising it and saying why it is relevant, so the reader can appreciate how the NROY ensemble relates to it.

Additional remarks have been added in the introduction and model description regarding the GLAC3 global ice sheet chronology model to provide some more context. However, the work presented in Lecavalier and Tarasov, (2025) and in this study is the Antarctic component that

is applied in the GLAC3 model. The GLAC3 global ice sheet chronology and its applications to global sea-level change is best reserved for another publication.

"As part of the history-matching analysis, a sub-ensemble of AIS simulations are chosen to represent the Antarctic component and its respective uncertainties within the global ice sheet chronology GLAC3 for future analysis."

"However, upon completing the full transient simulation, a gravitationally self-consistent solution is computed using the AIS simulation as part of the global ice sheet chronology GLAC3."

**Line-by-line comments**

L29-34: This feels more like a sequence of bullet points written in-line instead of proper text. Please rewrite and give it a proper flow for the reader, as it is hard (even if still possible) to follow the implications of one to another

Corrected.

"A large variety of ice loading histories and Earth rheologies are evaluated against the available data. Data-model comparisons are shown against a subset of the AntICE2 database which directly constrains relative sea-level (RSL) change and GIA. This illustrated significant spatial variability in Antarctic RSL and GIA. The uncertainties affiliated with these inferences are large given the limited number of observational constraints which results in inferred RSL bounds with max/min ranges up to 150 m during the Holocene."

L37: Please add a comma after "that" so the sentence actually states that it was your study that adequately explored the uncertainties, and not the previous studies.

Corrected.

L58: There's an extra ":" at the end of the line

This preceeds a list of different GIA models.

L63: Is the author's last name really "A"? I could not find it in the reference list

Error with reference manager, it was corrected.

L128-129: "ensemble parameter controlled three shell viscosity structure": some hyphenation needs to be done here so the reader can properly understand what is going on…

Corrected.

L186: Either "Antarctica" or "the Antarctic"

Corrected.

L195: A full stop works better than a comma after "matching"

Corrected.

L263: It is not clear which criteria were used to choose the HVSS. What counts as "High Variance" in this subset?

This is specified in part 1 of this study (Lecavalier and Tarasov, 2025):

"Here we present the data‑model comparison of the full ensemble, the NROY AN3sig sub-ensemble, and a highvariance subset (HVSS) selection from the AN3sig subensemble, with the latter being integrated within the GLAC3 global ice sheet chronology for future analysis. A HVSS of 18 simulations was extracted from the NROY AN3sig subensemble to showcase some glaciologically self-consistent simulation results. The simulations that make up the HVSS were selected based on maximizing the normalized multidimensional distance between metrics and scores for simulations in the NROY sub-ensemble. A few reference simulations with minimized scores for key data types were also included in the HVSS, such as the overall best-scoring simulation, best-scoring simulation against ice core data, and best-scoring simulation for marine paleo extent data. The HVSS simulations are shown against the LIG and LGM metrics of interest in Fig. S9 in the Supplement."

L323: Please change "Although" for "However"

Corrected.

L397: There's an extra "is" that does not make sense in this sentence

Corrected.

L438-440: It would be useful for the reader if this sentence was discussed more in terms of climate than "degrees of freedom", i.e., what kind of atmospheric, ocean, and basal conditions not captured in GSM would be necessary to fit the vertical motion estimates at sites 8426 and 8502?

We deem revisions are unnecessary since several factors could help address remaining data-model discrepancies, we can't point to a single aspect but it can be attributed to the limited range of forcings, processes, and feedback in a given region based on the existing degrees of freedom represented in the model through its ensemble parameters and boundary conditions. It is a non-unique problem to fit GPS and RSL data, therefore it is more important to talk about the range of scenarios a model can produce rather than simply state that we needed more precipitation over a site to increase the initial loading for an enventual unloading event since many scenarios can yield the same uplift rate.

L480: What is the difference between the minimum score and the joint minimum score? Is the GPS score not included in the former? If so, please clarify that in the text.

For a given simulation, it is scored against each data type in the AntICE2 database. The NROY sub-ensemble consists of simulations that are below the sigma thresholds on each of the data type scores in AntICE2. Within the NROY sub-ensemble, we identified the run (RefSim1) which has the minimum score across all the data type scores, we identified the run (RefSim4) with the lowest misfit score to the GPS data type, and we identified the run (RefSim5) which has the minimum score across the paleo data types (joint score based on the paleoH, paleoExt, paleoRSL data type scores). Rather than revise the text, the word "joint" was dropped from the text when speaking of data-model scores to be more succinct and avoid confusion.

L515: I believe it should be "reliance on three reference..."

Corrected.

L562: Here you state that the ensemble comprises 9,292 simulations, whereas in L16 and L255 it is stated 9,293. Please double check which one is correct

Corrected typo, should be 9293.

L571-579: I struggle to see how this paragraph fits in the Conclusions section. It reads much better without it, but I do understand that this relevant information. I'd suggest the authors to either rewrite it, or to move this to the previous section, making the appropriate changes so it fits in the text. This is related to my general comment #4

Restructured the conclusion to improve flow.

"In this study a sub-ensemble of Antarctic GIA inferences is presented based on a history-matching analysis of the GSM against the AntICE2 database. The fully coupled glaciological and GIA model was used to generate a full ensemble consisting of 9,293 Antarctic simulations spanning the last 2 glacial cycles. BANNs were trained to emulate the GSM for rapid exploration of the parameter space via MCMC sampling. Simulation results were scored against past relative sea level, PD vertical land motion, past ice extent, past ice thickness, borehole temperature profiles, PD geometry and surface velocity. The scores were used in the history-matching analysis to rule out simulations that were inconsistent with the data given observational and structural uncertainties, thereby a NROY sub-ensemble (N=82) that bound past and present GIA and sea-level change was generated.

Given that our history matching accounts for data-system and system-model uncertainties to a much deeper extent than any previous AIS study, the NROY sub-ensemble provides the most credible bounds to date on actual Antarctic GIA. As such, our analysis demonstrates that previous Antarctic GIA studies have underestimated the viscous deformation contribution to PD uplift rates due to past ice sheet changes across several key regions. This is particularly the case in the Amundsen sector, an area currently undergoing significant mass loss, which has a large range of viable PD GIA estimates. Our NROY set of chronologies will therefore facilitate more accurate inference of the PD mass balance of the AIS, including for vulnerable marine-based regions.

The NROY sub-ensemble of AIS results represent a collection of not-ruled-out-yet Antarctic components for the global GLAC3 ice sheet chronology. The NROY sub-ensemble AIS chronologies represent the Antarctic component in the GLAC global ice sheet chronology. This research enables the upcoming evaluation of global RSL predictions and the Antarctic far-field sea-level contributions during the last interglacial, LGM, and deglacial melt water pulses. The AIS NROY sub-ensemble chronologies are constrained by near-field observations. Evaluating the updated global ice sheet chronology against far-field RSL observations would in turn constrain the AIS NROY sub-ensemble by said far-field data, potentially ruling out additional AIS simulations that are currently in the NROY sub-ensemble. This future work could leverage 3D Earth GIA models to formally evaluate lateral Earth structure and its impact on far-field and near-field RSL predictions."

**Figures**

Figure 1: Please add to the caption what the abbreviations in the legend mean (paleoExt, paleoH, paleoRSL). In the text, only paleoH is explained

Corrected.

Figure 4 and all others in similar style: It looks like the grounding line shown is that of present day. I would recommend changing to that of one of the reference simulations, so the figures can better illustrate the solid-Earth response to changes in ice loading/unloading

We show the present day grounding line to georeference key features relative to present. It enables a better comparison across figures since the individual NROY sub-ensemble simulations exhibits a wide range of present day grounding line differences relative to present day which would likely confuse the reader. Additionally, it would be misleading to show ensemble mean, min, max, and 2sig ranges alongside reference simulation results since individual runs are glaciologically self consistent and ensemble statistics are not, so a proper unloading attribution like you describe would not be possible.

Figs 4 and S3: What is the significance of a RSL value where ice is grounded? If nothing, wouldn't it be clearer to mask out values where the ice is grounded in all ensemble members for each of the time slices? I would imagine this can be addressed in combination with a solution to my comment above.

RSL is the distance between the sea surface elevation and bedrock relative to present-day. The sea surface is located on a equipotential surface of the Earth's gravitational field i.e. geoid. Therefore, it has value to show inland RSL values since it also indicates the past geoid elevation inland which represents the past reference elevation with respect to sea level.

---

## Author Comment (AC3)

We would like to thank the reviewer for their comments, suggestions, and feedback. This response aims to address any comments raised by the reviewer. Our responses are embedded below and are shown in orange. Sections of text taken from the manuscript are shown in quotation marks "" while revisions/additions within these sections are underlined.

**Response to referree comments #1**

The work of Lecavalier and Tarasov assesses uncertainties in the solid Earth's response to loading and unloading of ice (and consequently the oceans) in the Antarctic. The authors show that there is a much larger uncertainty and different spatial patterns than previously estimated by the more popular Glacial Isostatic Adjustment (GIA) models, which are used to estimate the present-day AIS mass balance. Their assessment is done by history-matching a large ensemble of simulations using the Glacial Systems Model (GSM), producing a Not-Ruled-Out-Yet (NROY) subset that is further used as input to *adhoc* simulations of GIA using a more faithful solid-Earth model. The GSM ensemble is the same presented in a companion paper. They further use the NROY ensemble results to discuss implications for the climate and GIA (including when that means a limitation of the used forcings or solid-Earth models and parameter ranges).

Overall, the paper structure is mostly clear and easy to follow with just some points where the text could be improved, sometimes by rewriting confusing paragraphs, sometimes by clarifying some technical parts. Below I make some general remarks with suggestions to improve the overall state of the paper before it can be published, provide technical/editorial suggestions line by line, and finally comment on how to improve some of the figures presented.

I hope the authors find my comments useful, and I look forward to seeing a revised and improved version of this manuscript.

**General remarks**

1. The introduction adequately provides the background necessary to contextualise the paper, but it lacks a proper ending pointing the reader to what research question(s) the study aims to address. Please add a final paragraph or a couple of sentences framing how the present study fits into the picture provided, and what its goals are.

   Addressed with the following text:

   "This study is the second part of a two-part study, and should be considered in conjunction with part one which analyzed history matched ice sheet evolution and fits to non-GIA data constraints. (Lecavalier and Tarasov, 2025). Part two of this study presented below aims to quantify bounds on the evolution of Antarctic GIA. This is carried out via an approximate history-matching methodology that explicitly accounts for data and model uncertainties.. As part of the history-matching analysis presented in this 2 part study, a sub-ensemble of AIS simulations are chosen to represent the Antarctic component of the in progress GLAC3 (specifically version A denoted as GLAC3-A) global set of approximately history matched last glacial cycle ice sheet chronologies."

2. As per TC's guidelines, papers that are submitted or in prep (i.e., not yet available and without a DOI) cannot be cited. This needs to be rectified before the manuscript can be published. I suggest acting on it now instead of waiting for the same issue to be pointed out by the typesetting or copy-editing teams.

All the cited papers that were submitted or in prep listed in the manuscript now have a DOI. The manuscript has been updated with this information. The GSM description paper now has a permanent DOI at https://doi.org/10.5194/gmd-2024-175. Part 1 of this study is published: https://doi.org/10.5194/tc-19-919-2025. The Tarasov & Goldstein (2021) study can be found at: https://doi.org/10.5194/cp-2021-145.

3. Considering I am no GIA expert, and that this paper is likely targeted at paleoglaciologists and ice sheet modellers as well, I personally feel that the GIA model description part is quite confusing and a bit unstructured, and could be improved. It would be beneficial to this manuscript if the authors invested some time in improving the flow of the model description section (especially the last two paragraphs), rearranging some of the sentences to make the sequence of information presented more logical (e.g., not going back and forth between the GIA and ice sheet components) and adding some clarifications to the more technical terms (e.g., PREM structure). I believe such changes would provide a much better context for the results, and aid the non-GIA experts who would likely be interested in this paper.

The model description section was heavily revised to address this point though we still retain a separate paragraph describing the ensemble parameters given the central role they play in history matching:

"The GSM consists of comprehensive ice dynamic, climate forcing, and glacial isostatic components which are described in Tarasov et al. (2025) and Lecavalier and Tarasov (2025). To summarize the GSM includes: Hybrid SIA-SSA ice physics; with a subgrid grounding line ice flux parameterization; dual power basal drag for hard and soft bed sliding; ice shelf hydro-fracturing and restricted ice cliff failure; ocean temperature dependent sub ice shelf melt; subgrid ice shelf pinning point scheme; and a climate forcing representation with 14 ensemble parameters and fully coupled 2D energy balance climate model. An illustration showing the key components of the GSM is found in Figure S1 of Lecavalier and Tarasov (2025).
GIA models simulate the response of the solid Earth due to present and past changes in surface loading from the redistribution of ice, water, and mantle material. The two primary inputs to a GIA model are a global ice chronology and the Earth rheology. The GIA model products will henceforth be referred to as GIA inferences which include past and PD bedrock deformation, geoid and RSL estimates. The GSM is fully coupled to a GIA model of sea-level change based on a self-gravitating viscoelastic solid-Earth model which calculates GIA due to the redistribution of surface ice and ocean loads (Tarasov and Peltier, 1997) using a pseudo spectral solution for a spherically symmetric Earth rheology (Mitrovica and Peltier, 1991). The Earth model rheology has a standard Preliminary Reference Earth Model (PREM) density structure (Dziewonski and Anderson, 1981) which defines the radial elastic structure. The density structure is depth parameterized by volume

averaging the values into shells with thickness of 10.5 km in the crust and 25 km in the mantle. Three ensemble parameters specify the lithospheric thickness and viscosity of the upper and lower mantles. The lithospheric thickness, upper mantle viscosity, and lower mantle viscosity can respectively vary between 46 to 146 km, $0.1 \cdot 10^{21}$ to $5 \cdot 10^{21}$ Pa·s, and $1 \cdot 10^{21}$ to $90 \cdot 10^{21}$ Pa·s. The GIA component shares many similarities to that used in Whitehouse et al. (2012b) for post-processing modelled ice sheet chronologies but in contrast to the GSM, their ice sheet model did not have this component coupled (but instead used a simple local relaxation response parametrization). Given typical GIA response timescales, the GIA calculations are computed every 100 simulation years. To minimize the considerable computational cost of solving for a complete gravitationally self-consistent solution coupled with an ice sheet model (Gomez et al., 2010, 2013), a zeroth order geoidal approximation is used to account for the gravitational deflection of the sea surface. This approximation sums all ice sheet contributions to the local geoidal deflection from the global mean as detailed in Tarasov et al. (2025). However, upon completing the simulation, a gravitationally self-consistent solution is computed using the AIS simulation in combination with interim GLAC3 chronologies for the other last glacial cycle ice sheets (e.g. Tarasov et al., 2012; Kageyama et al., 2017; Kierulf et al., 2021) as per the methodology of Mitrovica and Peltier (1991). The complete solutions are those that are compared against the GPS and RSL observations in Section 4. The full continental scale transient Antarctic simulations over 205 ka have a 40 by 40 km horizontal resolution with the full sea-level solution having a spherical harmonic degree and order of 512.

As detailed in the accompanying study (Lecavalier and Tarasov, 2025), the Antarctic configuration of the GSM consists of 38 ensemble parameters. From the total 38 ensemble parameters, 10 are associated with ice dynamics (ice deformation, basal sliding), 11 with ice-ocean interactions (calving, sub-ice-shelf melt), 14 with ice-atmosphere interactions (atmospheric climate forcing), 3 with ice-solid Earth interactions (solid Earth rheology) and shown in Table 1 of Lecavalier and Tarasov (2025). Some of these ensemble parameters are applied to blend climate forcing schemes (parameterized PD climatologies, glacial index PMIP3 LGM climatologies, coupled energy balance climate model) to explore a wide range of plausible climate histories (Lecavalier and Tarasov, 2025; Tarasov et al., 2025). This represents the most comprehensive exploration of parametric uncertainties across the entire Antarctic glacial system of any study to date. A given simulation is defined by a parameter vector which consists of chosen values for each ensemble parameter. In this study, the GSM simulates AIS changes over the last 2 glacial cycles to minimize initialization uncertainties propagating into the last interglacial start of our history-matching interval. The GSM relies on several eustatic global sea-level forcing time series (e.g. Lambeck et al., 2014; Lisiecki and Raymo, 2005) when performing joint ice sheet and GIA calculations."

4. The authors offload most of the explanation regarding scoring the simulations to two other papers: One that is "in prep", and another that is an exceedingly lengthy pre-print which was never accepted for publication. The "in prep" manuscript is provided as part of the review process, which is much appreciated (I actually found it very interesting and look forward to seeing it eventually published). Still, it is very much in preparation, and I could only get a general grasp of how the scoring was done. Considering that details

regarding the scoring are not the focus of the manuscript under review, and "in prep" manuscripts cannot be cited, I would only ask that the authors explain slightly better why NROY simulations (or the entire ensemble, actually) do not bracket some observations, as evident in Figs 2 and 3. Is it because by choosing e.g., 3.5 or 4sigma means the "allowed variability" is actually larger than the ensemble variability itself? And what does the sigma refer to? Is it simply the standard deviation of the metric(s) being shown in the graphs?

The text was revised to include the following:

"Our implausibility threshold for inconsistency is a simulation-data misfit score component value of between 3-σ and 4-σ of the total uncertainty:

$$I = \frac{|d_i - E - \mu_{int} - \mu_{ext}|}{\sigma_{em} + \sigma_{struct} + \sigma_{obs}} \qquad [1]$$

The implausiblity (I) includes the data-model residual ($d_i - E$) as well at the total internal and external discrepancy bias ($\mu_{int}, \mu_{ext}$) and all uncertainty sources: emulator ($\sigma_{em}$) structural ($\sigma_{struct}$), observational standard deviation ($\sigma_{obs}$) (see Table S1 in Lecavalier and Tarasov, 2025 for values, and Tarasov and Goldstein, 2021 for motivation). The implausibility threshold for each data type ($\sigma$ in eq.1) is applied to the corresponding data-model score component. In other words, a NROY simulation must be NROY for each data type."

However, an implausibility threshold of 4sigma is consistently used for the RSL scores:

"given that the model struggles to bracket a few observations in this data class, which resulted in ruling out nearly all simulations if imposing a 3σ threshold across all data types."

At some RSL (9301, 9603, 9604) and GPS sites (8411, 8426, 8502, 8616), the data is not bracketed by the full ensemble, meaning any choice of implausibility threshold (3.5 or 4 sigma) would not yield any passing simulations that are consistent with the data (excluding considerably expanded structural uncertainty contributions). This suggests that the model is not exhibiting an adequate range of responses given the current accounting of model uncertainties in the GSM. This is discussed in Section 4.1 and 4.2:

"Given the full ensemble does not achieve the necessary amplitude needed to capture the observations at Windmill Island regardless of the range of Earth rheology considered in the history-matching analysis and Earth model sensitivity analysis, the discrepancy is likely due to the ice load chronology and/or a non-linear Earth rheology."

Also:

"This suggests the climate forcing or basal conditions may lack adequate regional degrees of freedom to produce a sufficiently larger ice load in the region, and subsequent deglaciation timing to reach the observed sea-level high-stand in the paleo RSL data."

However, at other RSL (9401) and GPS sites (8406, 8405, 8701), the full ensemble brackets the sea-level high stand or vertical land motion but upon imposing the implausibility thresholds, the NROY simulations fail to achieve the appropriate amplitude (e.g. reach the suggest 30 m sea-level highstand at site 9401). This suggest that the model is able to produce the adequate range of responses but upon imposing the implausiblity thresholds, all simulations that showed the adequate response failed to be broadly consistent, within several standard deviations, across the entire AntICE2 database. This is discussed in Section 4.1 and 4.2:

"The NROY sub-ensemble brackets the sites except for the highest sea-level observations at Terra Nova Bay (9401). The region is topographically complex with subgrid valley glaciers that are poorly resolved in the GSM. This is a recurring challenge performing a data-model comparison to paleoH data in the region which can manifest in an inaccurate ice unloading history. Moreover, this region of the Transantarctic Mountains has an anomalous low viscosity zone in the upper mantle which has consequences on the viscous response to past load changes (Whitehouse et al., 2019). The NROY HVSS Earth model sensitivity analysis demonstrates that by lowering the upper mantle viscosity in this region, a more rapid viscous response to ice unloading can reach the peak observed RSL at Terra Nova Bay."

"Tier-1 and tier-2 GPS observations that are not bracketed by the simulations tend to misfit both the full ensemble and NROY sub-ensemble at 3 distinct sites 8426, 8504, and 8502 in or near the Amundsen Sea sector. The Amundsen Sea sector has an anomalous low viscosity zone in the upper mantle which is not differentiated in the spherically symmetric GIA model. The HVSS Earth model sensitivity analysis does demonstrate that by considering a low upper mantle viscosity, elastic-corrected GPS predictions are captured at site 8406, 8411, 8504, 8616, and 8701 (Fig. S2) which are regions with inferred anomalously low viscosity structure. As the upper mantle viscosity is decreased by several orders of magnitude, this can significantly increase or decrease the amplitude of the GIA response depending on the temporal proximity of the unloading event. None of the HVSS Earth model sensitivity simulations capture the GPS bedrock trends (-4 mm/yr at site 8426, and 19 mm/yr at site 8502; Fig. S2), even though such amplitudes are attainable at other sites. Alternatively, the elastic corrections applied to the GPS data could be underestimated, particularly given their full uncertainties are ill-defined with its limited reliance on the input contemporary mass balance estimates (Martín-Español et al., 2016; Sasgen et al., 2017). Negative vertical land motion at 8426 suggests regional loading not represented by the elastic correction and/or GSM simulations. This could be attributed to increase precipitation or ice being advected to the region which thickens the ice column in the late Holocene. Conversely, 8502 with its exceedingly high elastic-corrected uplift rate indicates significant mass loss in the late Holocene. A large quantity of regional ice mass loss can be linked to ocean forcing and margin retreat of marine-based ice overlying soft till during the late Holocene. This suggests that the GSM might have insufficient degrees of freedom in the regional climate forcing and basal environment to produce a sufficiently late ice load scenario to reconcile these remaining discrepancies."

5. The GLAC3 chronology comes totally out of the blue, being mentioned only in the abstract and conclusions. All I can gather is that it stems from the NROY ensemble, but no other context is provided. It would be worth contextualising it and saying why it is relevant, so the reader can appreciate how the NROY ensemble relates to it.

Additional remarks have been added in the text to provide some more GLAC3 context:

"This yielded a NROY sub-ensemble of simulations consisting of 82-members that approximately bound past and present GIA and sea-level change given uncertainties across the entire glacial system. The NROY Antarctic ice sheet chronologies and associated Earth viscosity models represent the Antarctic component of the "GLAC3-A" set of global ice sheet chronologies over the last glacial cycle."

"Part two of this study presented below aims to quantify bounds on the evolution of Antarctic GIA. This is carried out via an approximate history-matching methodology that explicitly accounts for data and model uncertainties.. As part of the history-matching analysis presented in this 2 part study, a sub-ensemble of AIS simulations are chosen to represent the Antarctic component of the in progress GLAC3 (specifically version A denoted as GLAC3-A) global set of approximately history matched last glacial cycle ice sheet chronologies."

"Given typical GIA response timescales, the GIA calculations are computed every 100 simulation years. To minimize the considerable computational cost of solving for a complete gravitationally self-consistent solution coupled with an ice sheet model (Gomez et al., 2010, 2013), a zeroth order geoidal approximation is used to account for the gravitational deflection of the sea surface. This approximation sums all ice sheet contributions to the local geoidal deflection from the global mean as detailed in Tarasov et al. (2025). However, upon completing the simulation, a gravitationally self-consistent solution is computed using the AIS simulation in combination with interim GLAC3 chronologies for the other last glacial cycle ice sheets (e.g. Tarasov et al., 2012; Kageyama et al., 2017; Kierulf et al., 2021) as per the methodology of Mitrovica and Peltier (1991). The complete solutions are those that are compared against the GPS and RSL observations in Section 4."

**Line-by-line comments**

L29-34: This feels more like a sequence of bullet points written in-line instead of proper text. Please rewrite and give it a proper flow for the reader, as it is hard (even if still possible) to follow the implications of one to another

Addressed with the following revisions:

"This displays significant spatial variability in Antarctic GIA. The limited number of observational constraints contributes to wide inferred RSL bounds with max/min ranges up to 150 m during the Holocene."

L37: Please add a comma after "that" so the sentence actually states that it was your study that adequately explored the uncertainties, and not the previous studies.

Corrected.

L58: There's an extra ":" at the end of the line

This preceeds a list of different GIA models.

L63: Is the author's last name really "A"? I could not find it in the reference list

Error with reference manager, it was corrected.

L128-129: "ensemble parameter controlled three shell viscosity structure": some hyphenation needs to be done here so the reader can properly understand what is going on...

Corrected.

L186: Either "Antarctica" or "the Antarctic"

Corrected.

L195: A full stop works better than a comma after "matching"

Corrected.

L263: It is not clear which criteria were used to choose the HVSS. What counts as "High Variance" in this subset?

Addressed with the following revisions:

"A high variance 18 member subset (HVSS) of simulations were selected from the NROY sub-ensemble according to key metrics of interest, such as the AIS grounded ice volume during the Last Glacial Maximum (LGM) (Figure S7 in Lecavalier and Tarasov, 2025). Each metric of interest (last interglacial deficit and LGM excess relative to present) and AntICE2 data type scores were respectively normalized, and a simulation was chosen from the NROY subensemble to initialize the HVSS sampling (e.g. a NROY simulation with minimum total score across all data types). Each subsequent sample added to the HVSS is selected by identifying which simulation in the NROY subensemble maximizes the multidimensional distance (square root sum of squares) between all the normalized metrics and scores against the simulations already populating the HVSS. This method extracts a subset of simulations which exhibit a wide range of behaviours across the NROY sub-ensemble."

L323: Please change "Although" for "However"

Corrected.

L397: There's an extra "is" that does not make sense in this sentence

Corrected.

L438-440: It would be useful for the reader if this sentence was discussed more in terms of climate than "degrees of freedom", i.e., what kind of atmospheric, ocean, and basal conditions not captured in GSM would be necessary to fit the vertical motion estimates at sites 8426 and 8502?

Several factors could help address remaining data-model discrepancies, we can't point to a single aspect but it can be attributed to the limited range of forcings, processes, and feedback in a given region based on the existing degrees of freedom represented in the model through its ensemble parameters and boundary conditions. It is a non-unique problem to fit GPS and RSL data, therefore it is more important to talk about the range of scenarios a model can produce rather than simply state that we needed more precipitation over a site to increase the initial loading for an eventual unloading event since many scenarios can yield the same uplift rate. The text was partly revised to the following:

"Negative vertical land motion at 8426 suggests regional loading not represented by the elastic correction and/or GSM simulations. This could be attributed to increase precipitation or ice being advected to the region which thickens the ice column in the late Holocene. Conversely, 8502 with its exceedingly high elastic-corrected uplift rate indicates significant mass loss in the late Holocene. A large quantity of regional ice mass loss can be linked to ocean forcing and margin retreat of marine-based ice overlying soft till during the late Holocene. This suggests that the GSM might have insufficient degrees of freedom in the regional climate forcing and basal environment to produce a sufficiently late ice load scenario to reconcile these remaining discrepancies."

L480: What is the difference between the minimum score and the joint minimum score? Is the GPS score not included in the former? If so, please clarify that in the text.

For a given simulation, it is scored against each data type in the AntICE2 database. The NROY sub-ensemble consists of simulations that are below the sigma thresholds on each of the data type scores in AntICE2. Within the NROY sub-ensemble, we identified the run (RefSim1) which has the minimum of the maximum score across all primary data type scores, we identified the run (RefSim4) with the lowest misfit score to the GPS data type, and we identified the run (RefSim5) which has the minimum of the maximum score across the paleo data types ( paleoH, paleoExt, and paleoRSL). The text was revised to clarify this point. Additionally, the word "joint" was dropped from the text when speaking of data-model scores to be more succinct and avoid confusion.

"The reference simulations are all in the NROY sub-ensemble, where RefSim1 has the minimum of the maximum score across all the primary data types in AntICE2, RefSim4 has the minimum GPS score, and RefSim5 has the minimum of the maximum score across all the paleo data types in AntICE2."

L515: I believe it should be "reliance on three reference..."

Corrected.

L562: Here you state that the ensemble comprises 9,292 simulations, whereas in L16 and L255 it is stated 9,293. Please double check which one is correct

Corrected typo, should be 9293.

L571-579: I struggle to see how this paragraph fits in the Conclusions section. It reads much better without it, but I do understand that this relevant information. I'd suggest the authors to either rewrite it, or to move this to the previous section, making the appropriate changes so it fits in the text. This is related to my general comment #4

As recommended, the conclusion was restructured to improve flow.

"In this study a sub-ensemble of Antarctic GIA inferences is presented based on a history-matching analysis of the GSM against the AntICE2 database. The fully coupled glaciological and GIA model was used to generate a full ensemble consisting of 9,293 Antarctic simulations spanning the last 2 glacial cycles. BANNs were trained to emulate the GSM for rapid exploration of the parameter space via MCMC sampling. Simulation results were scored against past relative sea level, PD vertical land motion, past ice extent, past ice thickness, borehole temperature profiles, PD geometry and surface velocity. The full ensemble of simulations broadly brackets the AntICE2 database with a few outstanding data-model discrepancies likely attributed to model resolution, insufficiently climate forcing degrees of freedom for certain sectors, and insufficient accounting for uncertainties in the basal environment. In particular, this manifest in a few outstanding data-model discrepancies in regions with inadequately resolved complex topography such as the Transantarctic Mountains or regions with likely many subgrid pinning points that can help stabilize an ice shelf and grounding line. The scores were used in the history-matching analysis to rule out simulations that were inconsistent with the data given observational and structural uncertainties, thereby a NROY sub-ensemble (N=82) that bounds past and present GIA and sea-level change was generated.

Given that our history matching accounts for data-system and system-model uncertainties to a much deeper extent than any previous AIS study, the NROY sub-ensemble provides the most credible bounds to date on actual Antarctic GIA and last glacial cycle ice sheet evolution. As such, our analysis demonstrates that previous Antarctic GIA studies have underestimated the viscous deformation contribution to PD uplift rates due to past ice sheet changes across several key regions. This is particularly the case in the Amundsen sector, an area currently undergoing significant mass loss, which has a large range of viable PD GIA estimates. Our NROY set of chronologies will therefore facilitate more accurate inference of the PD mass balance of the AIS, including for vulnerable marine-based regions.

The NROY sub-ensemble of AIS results represent a collection of not-ruled-out-yet Antarctic components for the in progress global GLAC3 set of last glacial cycle ice sheet chronologies. Future research will prioritize a history-matching analysis using a higher horizontal resolution Antarctic configuration of the GSM, the integration of additional observational constraints such as the age structure of the ice inferred from reflective isochrones in radiostratigraphic data, and a 3D Earth viscosity GIA emulator (Love et al., 2024) to better represent lateral Earth structure."

**Figures**

Figure 1: Please add to the caption what the abbreviations in the legend mean (paleoExt, paleoH, paleoRSL). In the text, only paleoH is explained

Corrected.

Figure 4 and all others in similar style: It looks like the grounding line shown is that of present day. I would recommend changing to that of one of the reference simulations, so the figures can better illustrate the solid-Earth response to changes in ice loading/unloading

We show the present day grounding line to georeference key features relative to present. It enables a better comparison across figures since the individual NROY sub-ensemble simulations exhibits a wide range of present day grounding line differences relative to present day. We opted not to show ensemble mean, min, max, and 2sig ranges alongside a reference simulation result since individual runs are glaciologically self-consistent and ensemble statistics are not, so a proper unloading attribution like you describe would not be possible. However, the reference simulations in Figure 5d/e/f and 6d/e/f have been updated to also include their PD simulated GL.

Figs 4 and S3: What is the significance of a RSL value where ice is grounded? If nothing, wouldn't it be clearer to mask out values where the ice is grounded in all ensemble members for each of the time slices? I would imagine this can be addressed in combination with a solution to my comment above.

RSL is the distance between the sea surface elevation and bedrock relative to present-day. The sea surface is located on a equipotential surface of the Earth's gravitational field i.e. geoid. Therefore, it has value to show inland RSL values since it also indicates the past geoid elevation inland which represents the past reference elevation with respect to sea level.

---

## Author Comment (AC4)

We would like to thank the reviewer for their comments, suggestions, and feedback. This response aims to address any comments raised by the reviewer. Our responses are embedded below and are shown in orange. Sections of text taken from the manuscript are shown in quotation marks "" while revisions/additions within these sections are underlined.

**Response to referree comments #2**

This study presents an ensemble of coupled ice sheet–2D Glacial Isostatic Adjustment (GIA) simulations applied to the Antarctic Ice Sheet over the last two glacial cycles. The authors conducted 9,293 simulations and employs history matching techniques using Markov Chain Monte Carlo (MCMC) sampling and Bayesian Artificial Neural Networks (BANNs) to efficiently explore parameter space and refine model estimates. They selected a sub-ensemble of 82 simulations that best align with the bounds of past and present GIA and relative sea-level (RSL) observations. The selected sub-ensemble aims to provide improved constraints on past and present ice sheet evolution and the associated GIA response, contributing valuable insights into uncertainties in Antarctic mass balance assessments. The results indicate that the uncertainty in present-day GIA is greater than previously estimated by IJ05_R2 (Ivins and James, 2005), W12a (Whitehouse et al., 2012b), and ICE-6G_D (Peltier et al., 2015), particularly in the Amundsen Sea Embayment.

**General Remarks**

This study holds significant value for the scientific community. The analysis is well executed, and the figures are effectively presented. However, the methodology section is too concise and the section often refers to manuscripts in preparation and a preprint. Therefore, additional methodological details should be described within this manuscript. The interpretation of specific areas requiring additional explanation are outlined in the detailed comments below.

This manuscript is already lengthy and builds on part 1, hence why we rely on citations and opted to exclude an exhaustive model description and data scoring methodology section. The first part of this study is published and includes an abbreviateddescription of the history matching scoring and sigma thresholds in the methodology section (https://doi.org/10.5194/tc-19-919-2025). There are a variety of approaches to data-model scoring (e.g. Briggs et al., 2013; Ely et al., 2019) and the one applied in this study is broadly described in Tarasov & Goldstein (2019) (https://doi.org/10.5194/cp-2021-145). We are fine to have this paper not accepted until the history-matching methodology egusphere preprint becomes publicly available. Moreover, the GSM description paper is now available (https://doi.org/10.5194/gmd-2024-175). For these reasons, a mention in the introduction now explicitly states:

"This study is the second part of a two-part study, and should be considered in conjunction with part one for a complete understanding of the research (Lecavalier and Tarasov, 2025)."

The sensitivity test often cited to explain differences between the model ensemble and the dataset examines the effect of lateral Earth structure. However, using 2D structures with a relatively low mantle viscosity results in significantly different ice sheet evolution, bedrock uplift and sea level change than using 3D Earth structures (Gomez et al., 2018; van Calcar et al., 2023). Additionally,

the test employs an uncoupled GIA model, where lower viscosity affects ice sheet dynamics differently than in a coupled model, as the stabilizing GIA effect is absent. This likely leads to an overestimation of uplift between 15 and 5 ka due to stronger ice mass loss in the uncoupled model compared to a coupled model. Since the test does not fully assess the impact of lateral structure in the original coupled model ensemble but rather the effect of globally lower viscosity in an uncoupled model, its limitations should be acknowledged. While coupled 3D GIA–ice sheet simulations are not expected, the methods section should clarify these constraints, and the results should discuss potential uplift overestimation and its implications.

These limitations are explicitly stated in section 3 and 4 for the simulations that are part of the sensitivity analysis.

"The ice load chronologies from the 18 members of the NROY sub-ensemble HVSS were subject to repeated GIA post-processing over a range of Earth models. This involved applying each individual ice load chronology in the HVSS as input to the gravitationally self-consistent sea level solver using alternate Earth models. Specifically, the lithospheric thickness and upper mantle viscosity was progressively decreased from 146 km and $5 \cdot 10^{21}$ Pa·s to 46 km and $5 \cdot 10^{18}$ Pa·s (146, 120, 96, 71, to 46 km; $5 \cdot 10^{21}$, $5 \cdot 10^{20}$, $5 \cdot 10^{19}$, to $5 \cdot 10^{18}$ Pa·s), respectively, to evaluate the impact of an anomalously low upper mantle viscosity on isostasy (Fig. S1 and S2). This experimental design isolates the Earth model sensitivity at the cost of lost dynamical self-consistency between the ice history and Earth model."

"To better address this question will require a more involved specification of the structural uncertainty attributed to lateral Earth structure and/or a history matching analysis with a coupled 3D GIA Earth model. Moreover, the upper mantle viscosity sensitivity analysis shows that regions with a low viscosity zone can exhibit significant sensitivity to recent loading or unloading. Additionally, the sensitivity analysis does not consider coupled GIA feedbacks with these anomalously low upper mantle viscosities. Even though the NROY sub-ensemble is consistent with the entire AntICE2 database and represents a wide range of ice loading chronologies, 3D Earth GIA models could produce responses to ice loading that are not bracketed by the NROY sub-ensemble due to GIA feedbacks caused by lateral Earth structure. Thus, the NROY sub-ensemble GIA predictions likely underestimate the uplift and geoid rate bounds in specific regions with anomalous viscosity structure."

The history-matching analysis does perform a joint history matching of the AIS using an asynchronously coupled ice sheet and GIA model which results in a wide range of chronologies that are broadly consistent with the AntICE2 database. However, the sensitivity analysis is used as a secondary evaluation of remaining outstanding data-model discrepancies to assess the impact of a low viscosity lateral structure. Even though the simulations in the sensitivity analysis are not coupled, the ice load history being tested represent a wide range of loading scenarios.

A range of loading scenarios was applied in the sensitivity analysis to hopefully bracket the magnitude and timing of loading changes that would be represented by the GIA feedbacks in a fully coupled ice sheet and 3D Earth model. We recognize that the sensitivity analysis has limitations as stated in the text and the simulations aren't intended the reproduce the 3D Earth model GIA feedbacks but it represents an important evaluation that attempts to isolate the impact

of lateral structure on remaining data-model misfits. Three sentences were added (see above edits) to emphasize the lack of GIA feedback that might produce loading changes that are not bracketed by the full ensemble.

Last, the conclusion now consist of a short summary of what has been done and of recommendations for future work. This section would be improved by including more detail on the performance of the NROY subset and the 2sigma subset. Conclude which subset performs best in which region and the corresponding uncertainty ranges.

This is quite nuanced because the NROY sub-ensemble of simulations was approximately history matched against the entire AntICE2 database. Therefore, a given simulation can misfit paleoRSL data but achieve a good fit to paleoExt data. There are direct trade-offs when maximizing fits to only one data type over another. It is not as simple as stating the full ensemble has outstanding data-model misfits in a given region since typically it achieves a decent fit to several data types with the exception of some data. It is best framed in terms of the model limitations that might manifest in outstanding discrepancies rather than identifying regions where the NROY sub-ensemble struggles with a subset of data. For these reasons, this discussion is best left to section 4 results. Text was added to the conclusion to specify which model limitations are generally attributed to the remaining outstanding discrepancies and which regions are most impacted.

"The full ensemble of simulations broadly brackets the AntICE2 database with a few outstanding data-model discrepancies likely attributed to model resolution limitations, insufficiently climate forcing degrees of freedom for certain sectors, and insufficient accounting for uncertainties in the basal environment. In particular, this manifest in a few outstanding data-model discrepancies in regions with inadequately resolved complex topography such as the Transantarctic Mountains or regions with many subgrid pinning points that can help stabilize an ice shelf and grounding line."

**Line-by-Line Comments**

**L123:** The expanded climate forcing scenarios are known to be of high influence on the ice sheet model and is later in this manuscript mentioned as an important source of uncertainty. It would therefore be useful mention which expanded climate forcing scenarios have been applied.

This study aims to focus on Antarctic GIA and opted to instead cite the GSM description paper which offers a more comprehensive description (Section 2.10 of Tarasov et al., 2025; https://doi.org/10.5194/gmd-2024-175). Additions to the text were made to address this comment:

"From the total 38 ensemble parameters, 10 are associated with ice dynamics (ice deformation, basal sliding), 11 with ice-ocean interactions (calving, sub-ice-shelf melt), 14 with ice-atmosphere interactions (atmospheric climate forcing), 3 with ice-solid Earth interactions (solid Earth rheology) and shown in Table 1 of Lecavalier and Tarasov (2025). Some of these ensemble parameters are applied to blend climate forcing schemes (parameterized PD climatologies, glacial index PMIP3 LGM climatologies, coupled energy balance climate model) to explore a wide range of plausible climate histories (Lecavalier and Tarasov, 2025; Tarasov et al., 2025)."

**L126-128:** Please explain how the PREM density structure is applied to the layers of in the model.

The Preliminary Reference Earth Model density structure defines the radial elastic structure. This is standard practice for 1D GIA models with spherically symmetry earth rheology. The text was expanded as follows:

"The Earth model rheology has a standard Preliminary Reference Earth Model (PREM) density structure (Dziewonski and Anderson, 1981) which defines the radial elastic structure. The density structure is depth parameterized by volume averaging the values into shells with thickness of 10.5 km in the crust and 25 km in the mantle. Moreover, an ensemble parameter-controlled three-shell viscosity structure defined by … "

**L130-131:** Please mention explicitly what the GIA component in post-processing modelled ice sheet chronologies is.

We do not entirely understand the request. The text was party revised to hopefully provide clarity:

"The GSM is fully coupled to a GIA model of sea-level change based on a self-gravitating viscoelastic solid-Earth model which calculates GIA due to the redistribution of surface ice and ocean loads (Tarasov and Peltier, 1997) using a pseudo spectral solution for a spherically symmetric Earth rheology (Mitrovica and Peltier, 1991)."

What is described above represents the GIA component in the GSM. However, during a transient ice sheet simulation the GIA calculations do not compute the full gravitationally self-consistent solution. Only at the end of a transient simulation, a post-processing step computes the full gravitationally self-consistent solution as described:

"Given typical GIA response timescales, the GIA calculations are computed every 100 simulation years. To minimize the considerable computational cost of solving for a complete gravitationally self-consistent solution coupled with an ice sheet model (Gomez et al., 2010, 2013), a zeroth order geoidal approximation is used to account for the gravitational deflection of the sea surface. This approximation sums all ice sheet contributions to the local geoidal deflection from the global mean as detailed in Tarasov et al. (2025). However, upon completing the simulation, a gravitationally self-consistent solution is computed using the AIS simulation in combination with interim GLAC3 chronologies for the other last glacial cycle ice sheets (e.g. Tarasov et al., 2012; Kageyama et al., 2017; Kierulf et al., 2021) as per the methodology of Mitrovica and Peltier (1991)."

Or perhaps there is confusion associated with the fact that Whitehouse et al 2012 did not have coupled GIA in their ice sheet model. Just in case, we've added

"The GIA component shares many similarities to that used in Whitehouse et al. (2012b) for post-processing modelled ice sheet chronologies but in contrast to the GSM, their ice sheet model did not have this component coupled (but instead used a simple local relaxation response parametrization)."

which clarifies that their GIA calculations are conducted after their ice sheet simulation finishes as part of a post-processing step.

**L135-136:** Explain in detail what is meant by zeroth order geoidal approximation and how this is applied.

The following revisions were implemented to address this comment:

"Given typical GIA response timescales, the GIA calculations are computed every 100 simulation years. To minimize the considerable computational cost of solving for a complete gravitationally self-consistent solution coupled with an ice sheet model (Gomez et al., 2010, 2013), a zeroth order geoidal approximation is used to account for the gravitational deflection of the sea surface. This approximation sums all ice sheet contributions to the local geoidal deflection from the global mean (as per the imposed eustatic sea level chronology) as detailed in Tarasov et al. (2025)."

It is a method that accounts for approximate geoidal changes due to GIA and ice load changes by scaling spatial geoidal deflection according to ice sheet volume. The deflection fields are from previously history matched full gravitationally-self-consistent GIA solutions and the resultant deflections are then applied to the mean sea-level forcing. A detailed description along with a sample comparison against a complete sea level solution can be found Section 2.15 of Tarasov et al. (2025).

**L136-138:** At this point in the text, it is not clear what is meant by "full transient simulation" and why the simulations in section 4 are different from the simulations in the other sections. Please elaborate in the text which method is used for which simulations exactly. It would be useful to end the introduction with a short overview of which simulations are discussed in which section so that the reader has got an overview to which you can refer to lines 136-138.

This was corrected, instead here we only refer to the "simulation" since two sentences later we specify what is meant by a full transient simulation (205 ka to present). Effectively, all the simulations in the history-matching analysis are full transient simulations using the GSM. Except that the sensitivity analysis are subject to only GIA post processing with an alternate Earth model.

"The full continental scale transient Antarctic simulations over 205 ka have a 40 by 40 km horizontal resolution with the full sea-level solution having a spherical harmonic degree and order of 512."

**L148-149:** Please indicate why these ranges for viscosity and lithospheric thickness have been chosen and discuss the implications of this choice on the results. As mentioned later in the manuscript, it has been shown that the upper mantle viscosity can regionally be orders of magnitude lower than 0.1*10^21 pa s (e.g. Barletta et al., 2018). Furthermore, it has been shown that using a uniform upper mantle viscosity of, for example, 10^19 pa s better represents a laterally varying Earth structure than a uniform viscosity of 10^21 pa s because most ice mass changes have occurred over the West Antarctic Ice Sheet and the average viscosity of the West Antarctic Ice Sheet is an order of magnitude lower than 10^21 pa s (van Calcar et al. 2023).

Revisions to the text have been made to emphasize that future work should apply a broader range of upper mantle viscosities. The Antarctic continent represents a large region and given that we are using a spherically symmetric GIA model, the chosen range for the Earth viscosity ensemble parameters for the lithospheric thickness, upper mantle viscosity, and lower mantle viscosity is

between 46 to 146 km, $0.1 \cdot 10^{21}$ to $5 \cdot 10^{21}$ Pa·s, and $1 \cdot 10^{21}$ to $90 \cdot 10^{21}$ Pa·s, respectively. The literature suggests this is a sufficiently wide range for the studied domain size (e.g. Whitehouse et al., 2019). Indeed there are small regions with anomalous viscosity deviations that exceed this range but our focus was on full continental scale simulation. The low viscosity Earth model GIA simulations (upper mantle viscosities as low as $5 \cdot 10^{18}$ Pa s) were relegated to the sensitivity analysis to quantify their uncoupled impact.

"A future history-matching analysis should apply a broader range of upper mantle viscosity Earth models to include values down to $10^{19}$ pa·s (van der Wal et al., 2015) as applied in our sensitivity analysis given that WAIS rests atop several anomalously low viscosity zones and this region experienced the greatest mass change since the last interglacial. However, it remains unclear to which degree it is appropriate to use such low viscosity values for glacial cycle simulations given that model-based support for use of such a low viscosity in visco-elastic models coupled to ice sheet models without lateral earth viscosity variation has only been shown for contemporary load change contexts (van der Wal et al., 2015). "

Whitehouse, P.L., Gomez, N., King, M.A. and Wiens, D.A., 2019. Solid Earth change and the evolution of the Antarctic Ice Sheet. Nature communications, 10(1), p.503.

van der Wal, W., Whitehouse, P.L. and Schrama, E.J.: Effect of GIA models with 3D composite mantle viscosity on GRACE mass balance estimates for Antarctica. Earth and Planetary Science Letters, 414, pp.134-143, 2015.

**L212-216:** It is not clear how the 5% error estimate for RSL was chosen. Additionally, the range of bias error for present-day uplift rates needs further clarification. Is this range site-dependent, and if so, what parameters influence its variation?

The text was revised to provide clarity on this comment:

"This includes a 5% structural bias error for RSL and a +1 mm/yr and -0.5 mm/yr bias error for PD vertical uplift both due to the use of a spherically symmetric global 1D Earth rheology instead of a 3D Earth rheology. These values were chosen on the basis of discrepancies between corresponding results for regional 1D and 3D Earth rheology modelling of Fennoscandia (Whitehouse et al., 2006; Whitehouse, 2009). However, a similar analysis for Antarctica is lacking and needs to be addressed by the community."

**L220:** The concept of multi-million-point Markov Chain Monte Carlo (MCMC) sampling should be explained.

The following was included in the text to address this comment:

"MCMC sampling uses a sequence of dependent autocorrelated samples generated through a Markov chain to approximate high dimensional probability distributions. Each sampling step in each chain requires a comparison of GSM predictions against the constraint data."

**L220:** Please also mention how many simulations were included in the original Glacial Systems Model (GSM) ensemble.

These details are described later on in this section of this study:

"Previous ensembles of simulations were evaluated against the AntICE2 database and PD observations to verify that the observations are adequately bracketed by the GSM given uncertainties (Lecavalier and Tarasov, 2025)." … "These enhancements, along with broader parameterizations of marine basins and other model components, led to an increase in the number of ensemble parameters, process additions to the GSM, and revisions to certain boundary conditions. Leading up to the final waves of ensembles, over 30,000 model simulations were performed as part of previous experimentation, sensitivity analyses, Latin Hypercube, and Beta fit sampling of ensemble parameters. In the results section, we present the latest iterations of large-ensemble results based on the history-matching analysis which consists of the final 9,293 simulations."

The text was revised to improve the overall clarity on this topic:

"This final waves of ensembles (N=9,293) is henceforth referred to as the "full ensemble" (see Table 2 in Lecavalier and Tarasov, 2025)."

**L221-226:** For improved readability, consider moving the sentence "The BANN targets… for data-model comparison" to line 222 before "The BANNs were trained…"

Corrected.

**L226-227:** The term "MCMC converged sampling chains" requires further explanation. How was convergence assessed? Additionally, more context is needed on the role of the Bayesian Artificial Neural Network (BANN) architectures prior to this sentence.

We are not clear on what is meant by context on the role of the BANN architecture. The specifics on the internal structure of the BANN is well beyond the bounds of this submission. In case the confusion is in regards to BANN, we have added:

"A BANN emulator is a probabilistic surrogate model with a specific architecture of interconnected layers of artificial neurons that approximates the behaviour of a complex system by learning from an ensemble of simulation outputs."

Additions to the text have been made to clarify how MCMC convergence is assessed:

"In the high-dimensional non-linear glacial system, it is challenging to ensure that all not implausible parameter space sectors have been identified. Consequently, this study utilized hundreds of MCMC sampling chains that are initiated from widely dispersed points in the parameter space. A sub-sample of parameter vectors from the converged part of at least 100 MCMC sampling chains is in turn used to create an ensemble of GSM simulations. Convergence is assessed according to MCMC diagnostics as well from examining the evolution of a few of the state variables over the chain steps. (Neal, 2012)"

Neal, R.M.: Bayesian learning for neural networks (Vol. 118). Springer Science & Business Media, 2012.

**L245-247:** What measures have been taken to address the potential issue of overfitting? Could the discrepancies be due to deficiencies in the climate model rather than model overfitting? Please provide a discussion on overfitting control.

Overfitting occurs when forcing a model to fit data without (or with insufficient) accounting for uncertainties, thus effectively fitting the model to the noise in the data. History matching is therefore the complete opposite given its emphasis on complete uncertainty assessment/specification, lack of parameter tuning/optimization, and focus on ruling out model configurations that are clearly inconsistent with the chosen constraint data set. So we do not understand why overfitting is being raised here. If the referee's question is motivated by the dimension of the ensemble parameter space, aside from the above points, this dimension is still limited compared to the uncertainty and degrees of freedom of the ice and climate system over the last glacial cycle.

We are confused as to what is meant by discrepancies possibly being due to over-fitting as the contrary would ensue. The line context for this question is also strange, as the discussion in L245-7 is about inadequate fits to data constraints, overfitting would cause the opposite. As such, we see no point in modifying the relevant text.

 The discrepancies discussed in this part are likely due to a host of model limitations (including grid resolution, input data sets, process representation, ...). As we explicitly stated in the original submission, these discrepancies were reduced by adding processes, ensemble parameters, and updating some input boundary conditions. Compared to other paleo ice sheets, uncertainties in atmospheric climate forcing are much less critical (given the negligible amount of present-day surface melt), however uncertainties in the ocean temperature forcing and associated submarine melt are a challenge. Expanding the degrees of freedom in the ocean temperature forcing was therefore a significant part of the GSM updates carried out in this iterative process of history matching as stated:

"To address any persistent data-model discrepancies following history-matching waves, the GSM underwent major model updates. A key refinement involved expanding the degrees of freedom in the ocean temperature forcing. Sub-ice-shelf mass balance in the GSM is computed using an ocean-temperature-dependent parameterization at the ice–ocean interface (Tarasov et al., 2025), encompassing the ice front, grounding line, and sub-shelf regions. Ocean temperature forcing is derived from transient TraCE-21ka simulations (He, 2011), bias-corrected using PD ECCO reanalysis data (Fukumori et al., 2018). For periods prior to 21 ka, a glacial index scheme is applied to the bias-corrected TraCE-21ka outputs. The temperature field beneath ice shelves is extrapolated with a depth cut-off based on minimum sill height to account for deeper continental shelves. To avoid extrapolating TraCE-21ka temperatures under warmer-than-present conditions, particularly relevant for Last Interglacial simulations, a separate ensemble parameter (rToceanWrm) was introduced. This parameter scales glacial-index-derived atmospheric warming and adds it to the PD ocean climatology, leveraging the empirical relationship between Antarctic $\delta^2H$ and mean ocean temperature (Shackleton et al., 2021). These enhancements, along with broader parameterizations of marine basins and other model components, led to an increase in the number of ensemble parameters, process additions to the GSM, and revisions to certain boundary

conditions. Leading up to the final waves of ensembles, over 30,000 model simulations were performed as part of previous experimentation, sensitivity analyses, Latin Hypercube, and Beta fit sampling of ensemble parameters."

**L248:** Clarify which previous ensembles of simulations are being referred to.

We've added a citation to part 1 of our study to make clear where the verification of ensemble bracketing of key data constraints was verified. Going beyond this would likely confuse the issue (what benefit would there be from cataloguing each of the 10 ensembles used in the course of this project?)

**L255:** The selection process for the 9,293 simulations is unclear. Additionally, where did the 30,000 model simulations originate? Are there references supporting this methodology? A step-by-step description of how simulations were filtered and refined would be beneficial.

This is detailed in part 1 of this study in Section 4 (Lecavalier and Tarasov, 2025), the ensembles are summarised in Table 2, the history matching data-type thresholds are listed in Table S1, and the history-matching methodology is diagramtically illustrated in Figure S8. For these reasons, no revisions to the text are necessary since the focus of this study is predominantly on Antarctic GIA.

**L263-267:** Define how "high variance" is measured in the subset. What parameters exhibit high variance? What criteria were used for selection? List all key metrics of interest and the minimum scores to which relevant data types.

The text was expanded to include some of this information to address this comment:

"Each metric of interest (LIG deficit and LGM excess relative to present) and AntICE2 data type scores were respectively normalized, and a simulation was chosen from the NROY subensemble to initialize the HVSS sampling (e.g. a NROY simulation with minimum total score across all data types). Each subsequent sample added to the HVSS is selected by identifying which simulation in the NROY subensemble maximizes the multidimensional distance (square root sum of squares) between all the normalized metrics and scores against the simulations already populating the HVSS. This method extracts a subset of simulations which exhibit a wide range of behaviours across the NROY sub-ensemble."

**L268:** Clarify how the repeated GIA post-processing has been done.

We want to provide some clarity on what is meant by GIA post-processing. We partly address this with the revisions made to address comments linked to L130-131. Regardless of whether an ice sheet model is coupled to a GIA model or not, the resulting ice load chronology can be used as input to a GIA model. This is what we mean by GIA post-processing, we apply an existing ice load history as input to do several GIA simulations using alternate Earth models, hence GIA post-processing. An additional sentence is added which clarifies this step:

"The ice load chronologies from the 18 members of the NROY sub-ensemble HVSS were subject to repeated GIA post-processing over a range of Earth models. This involved applying each individual ice load chronology in the HVSS as input to the gravitationally self-consistent sea level solver using alternate Earth models."

**L269:** Also include the upper limits for lithospheric thickness and upper mantle viscosity, as well as the step size used in the sensitivity analysis.

This is illustrated in the legend of both Figure S1 and S2. The text was revised to explicitly specify this information:

"Specifically, the lithospheric thickness and upper mantle viscosity was progressively decreased from 146 km and $5 \cdot 10^{21}$ Pa·s to 46 km and $5 \cdot 10^{18}$ Pa·s (146, 120, 96, 71, to 46 km; $5 \cdot 10^{21}$, $5 \cdot 10^{20}$, $5 \cdot 10^{19}$, to $5 \cdot 10^{18}$ Pa·s), respectively, to evaluate the impact of an anomalously low upper mantle viscosity on isostasy (Fig. S1 and S2)."

**L271:** Figures S1 and S2 are referenced but not discussed. A brief summary of their implications should be included in the main text to guide the reader.

No revisions are made given there are 8 references to Figure S1 and S2 that discuss the implication of these figures in the Results and Discussion section.

**L289-290:** Please elaborate on how GPS measurements are elastically corrected. Could this correction method contribute to the discrepancies in the model fit for West Antarctica? A brief discussion on the limitations of the elastic correction approach would be valuable.

Addressed with the following additions:

"With regards to the latter, the GPS measurements have to be corrected for the elastic response due to recent ice mass changes to isolate the viscous signal due to past load changes (Martín-Español et al., 2016; Sasgen et al., 2017). We rely on the validity of the elastic corrections which are dependent on the inferred contemporary ice load changes which have their own explicit and some ill-defined implicit uncertainties. Thus, only GPS data that is minimally impacted by a contemporary elastic signal are considered."

**L322:** Can the required upper mantle viscosity for data consistency in this region be quantified?

Partly, the challenge is that this is a joint ice and Earth model history-matching analysis. The sensitivity analysis attempts to evaluate if a lower upper mantle viscosity can address an outstanding misfit in the region. Simply because it does in this region does not imply that we have constrained the regional viscosity structure since this is a non-unique problem and we simply demonstrate that these alternate Earth models can help rectify the last remaining misfits at site 9101. The sensitivity analysis suggests that for a given loading history and Earth model pair, you might be able to address these final misfits with a GIA component that includes lateral variations in Earth viscosity. However, this does not necessary imply that the upper mantle viscosity in this region is $0.05 \cdot 10^{21}$ Pa·s. The following additions to the text were made:

"Therefore, lateral Earth structure that corresponds to a lower upper mantle viscosity can produce a RSL fall consistent with the data which corresponds to an upper mantle viscosity of $5 \cdot 10^{19}$ Pa·s. However, there is limited evidence of lateral structure in this region (Whitehouse et al., 2019). This suggests the unloading history and magnitude of ice loss could be responsible for the discrepancy with the sea-level high-stand data in this area."

**L323-324:** The manuscript lacks a clear definition of "lateral structure" and whether is or is not a lateral structure in a certain region. Lateral variations in viscosity depend on spatial scale, and the sensitivity of RSL measurements depends on the extent of ice (un)loading. For example, at the Syowa Coast, viscosity varies by one to two orders of magnitude over 300 km (e.g. Ivins et al., 2023). The inferred average viscosity of ~10^20 Pa·s for this region appears reasonable. Additionally, while inconsistencies could stem from climate forcing, the possibility of errors in the Earth structure model should not be dismissed.

An addition to the text was made to explicity quantify what we mean by lateral Earth structure.

"In this study, reference to lateral Earth structure in the upper mantle is broadly defined wherever there is a gradient in upper mantle viscosity that exceeds approximately one order of magnitude on 100 to 1,000 km length scales (e.g. Figure 4 from Whitehouse et al., 2019)."

Generally we refer to lateral Earth structure as motivation for considering alternate Earth models as part of the sensitivity analysis to explore its potential impact to isostasy. Even though some regions have considerable, limited, to no evidence of lateral Earth structure, we agree that we cannot dismiss this impact even though other sources of uncertainty has the potential to address remaining data-model discrepancies.

**L454-455:** Provide a clearer explanation of the distinction between nominal ranges and Gaussian confidence intervals.

The 2 sigma range for a Gaussian distribution is based on the 0.02275 and 0.97725 quantiles which represents the 95.45% interval. We apply these same quantiles on the ensemble results rather than the exact 95% interval, hence why we refer to them as the nominal 95% interval. At L285-287 we define the nominal intervals but revisions were made to provide clarity:

"The 2σ and 1σ ensemble ranges shown across several figures (e.g. Fig. 2-7) are the nominal 95% and 68% ensemble intervals based on the equivalent Gaussian quantiles (95.45 % = 97.725 - 2.275 % Gaussian 2σ quantiles and 68.268 % = 84.134 - 15.866 % Gaussian 1σ quantiles)."

**L455-456:** Briefly justify why the reader should consider the complete NROY sub-ensemble. How do the results of the full NROY sub-ensemble compare to those within the nominal 2σ range? A short discussion on the added insights from the complete sub-ensemble would be beneficial.

In the supplementary we show the min and max of the NROY sub-ensemble spatial fields, while in the main text we only show the 2sigma ranges. The strength of the history-matching analysis is to establish plausible bounds that are consistent with the data. However, when looking at spatial fields, the visualization can be dominated by a single outlier, hence why we show both in this study. The discussion in the text reflects those from the NROY sub-ensemble min/max and only mention the nominal 2sigma uncertainties with regards to spatial plots but for completeness include the spatial min/max figures in the supplementary section. We state the following in the manuscript which covers these points:

"History matching on its own does not produce a probability distribution of chronologies. Throughout this study, we present the NROY sub-ensemble nominal 2σ range since studies that apply GIA corrections are typically interested in accounting for nominal 2σ uncertainties. Moreover, by visualizing 2σ ranges one avoids any one outlier simulation from dominating the visualization. However, these nominal ranges should not be confused with traditional Gaussian confidence intervals and the reader is encouraged to also consider the complete NROY sub-ensemble min and max GIA and RSL ranges shown in Figure S3-S6. When evaluating the spatial RSL, uplift rates, and geoid NROY sub-ensemble plots (Figure 4, 5, 6, and 7), the min/max and -2σ/2σ plots differ primary by magnitude where the NROY sub-ensemble min is broadly more negative than those of the -2σ field (conversely true for the NROY sub-ensemble max and 2σ field). The difference between the min/max and -2σ/2σ plots illustrate the impact of edge case simulations that made it within the NROY sub-ensemble which can be considerable in certain regions (e.g. >10 m difference in WAIS RSL at 10 ka between the max and 2σ field as shown in Figure 4f and S3f)."

**L466-467:** The phrasing suggests that uncertainties in regional RSL measurements are large. However, given the relatively small error bars in Figure 2, the uncertainty appears to stem from the model fit to all Antarctic RSL measurements rather than the measurements themselves. The sentence should be reworded to emphasize model uncertainty sources, including the choice of climate and GIA models (2D instead of 3D).

We are indeed referencing the modelled RSL predictions based on the NROY sub-ensemble which is data-constrained by the AntICE2 database. Regions in the spatial plots of Figure 4 and S3 with a large min/max and 2sigma range represent regions that are poorly constrained by the data and/or large structural errors in the GSM. The text was revised to explicitly specify that we are discussing modelled RSL uncertainties:

"This emphasizes the large NROY RSL range due to competing observational constraints and structural errors in the entire glacial system."

**L469-471:** Provide examples of specific regions where this effect occurs. Do these correspond to areas with the largest RSL uncertainty ranges?

These regions have smaller modelled RSL uncertainties because they are physically constrained by the maximum ice load that can expand across the continental shelf. The text was revised to mention two examples:

"The maximum ice load was also physically constrained in some regions during the LGM because of a limited continental shelf, particularly along the Dronning Maud - Enderby Land and Wilkes – Victoria Land sectors. This directly limits the maximum possible ice load in certain regions which impacts the subsequent deglacial GIA response."

**L570:** GLAC3 is mentioned for the first time here. Please include an explanation in the method section.

We've expanded the text that discusses the GLAC3 model to provide some concise background information.

GLAC3 mentions in the manuscript:

Abstract

"This yielded a NROY sub-ensemble of simulations consisting of 82-members that approximately bound past and present GIA and sea-level change given uncertainties across the entire glacial system. The NROY Antarctic ice sheet chronologies and associated Earth viscosity models represent the Antarctic component of the "GLAC3-A" set of global ice sheet chronologies over the last glacial cycle."

Introduction

"Part two of this study presented below aims to quantify bounds on the evolution of Antarctic GIA. This is carried out via an approximate history-matching methodology that explicitly accounts for data and model uncertainties. As part of the history-matching analysis presented in this 2 part study, a sub-ensemble of AIS simulations are chosen to represent the Antarctic component of the in progress GLAC3 (specifically version A denoted as GLAC3-A) global set of approximately history matched last glacial cycle ice sheet chronologies."

Model description

"Given typical GIA response timescales, the GIA calculations are computed every 100 simulation years. To minimize the considerable computational cost of solving for a complete gravitationally self-consistent solution coupled with an ice sheet model (Gomez et al., 2010, 2013), a zeroth order geoidal approximation is used to account for the gravitational deflection of the sea surface. This approximation sums all ice sheet contributions to the local geoidal deflection from the global mean as detailed in Tarasov et al. (2025). However, upon completing the simulation, a gravitationally self-consistent solution is computed using the AIS simulation in combination with interim GLAC3 chronologies for the other last glacial cycle ice sheets (e.g. Tarasov et al., 2012; Kageyama et al., 2017; Kierulf et al., 2021) as per the methodology of Mitrovica and Peltier (1991). The complete solutions are those that are compared against the GPS and RSL observations in Section 4."

Conclusion

"The NROY sub-ensemble of AIS results represent a collection of not-ruled-out-yet Antarctic components for the in progress global GLAC3 set of last glacial cycle ice sheet chronologies."

**Figure 1:** The caption should briefly define the abbreviations used in the legend (e.g., paleoExt, paleoH, paleoRSL).

Corrected.

**Figure S1 & S2:** Ensure consistency between the legend and caption regarding viscosity values. The notation should be clarified, and redundant legends should be removed for clarity. The caption states that UMV varies from $5\cdot10^{21}$ to $0.005\cdot10^{21}$ Pa·s, but the legend lists values of 0.05 and 0.005, presumably referring to $5\cdot10^{19}$ and $5\cdot10^{18}$ Pa·s. Clarify this discrepancy and ensure consistency in notation.

The upper mantle viscosity (UMV) in the legend goes from $5 \cdot 10^{21}$ Pa·s (Ref Earth with UMV=5) to $0.005 \cdot 10^{21}$ Pa·s (RefEarth with LT=46 UMV=0.005) which is consistent with the caption.

**References** E. R. Ivins, W. van der Wal, D. A. Wiens, A. J. Lloyd, L. Caron, 2023. "Antarctic upper mantle rheology," The Geochemistry and Geophysics of the Antarctic Mantle, A. P. Martin, W. van der Wal.

**Citation**: https://doi.org/10.5194/egusphere-2024-3268-RC2

---

## Editor Decision (ED1)

[revised manuscript text omitted]

The interaction between the solid Earth and the AIS through GIA is dictated by the rate and magnitude of ice sheet changes, and Earth's rheological properties. However, beneath the Antarctic continent there are large variations in rheological properties of the mantle (Ritzwoller et al., 2001; Schaeffer and Lebedev, 2013; Heeszel et al., 2016; Shen et al., 2018; Lloyd et al., 2020). These rheological variations define the viscosity of the mantle and the subsequent relaxation timescales from a surface loading or unloading event. There are regions of apparently anomalously low upper mantle viscosities in the Antarctic Peninsula and Amundsen Sectors that experience a more rapid GIA response to mass loss relative to other Antarctic sectors (Nield et al., 2014; Barletta et al., 2018). This implies that the present-day (PD) viscous GIA signal in regions with low effective mantle viscosity is possibly more dominated by ice sheet changes over the last several millennia rather than early deglacial ice sheet mass loss. As such, any approach to past inference of AIS evolution should account for this lateral variation in effective Earth viscosity.

The contemporary mass balance of the Antarctic ice sheet is inferred using a variety of geodetic methods. Ice sheet changes can be monitored using satellite altimetry, radar imagery, optical imagery, and gravimetry (e.g. Mouginot et al., 2017; Gardner et al., 2018; Smith et al., 2020; Tapley et al., 2019; Velicogna et al., 2020; Sasgen et al., 2020). However, to infer the mass balance of the AIS using these methods, GIA estimates with meaningful uncertainty estimates are required (Shepherd et al., 2018; Otosaka et al., 2023). The Ice Sheet Mass Balance Inter-comparison Exercise (IMBIE) initially reconciled these various satellite methods (Shepherd et al., 2012, 2018) and has continued to provide mass balance estimates extending to 2020 (Otosaka et al., 2023). By combining these different mass balance inference methodologies, confidence in both contemporary mass balance and contributions to sea-level rise are enhanced. Uncertainties in Antarctic GIA dominate the contemporary mass balance confidence intervals. As of now, all these methods rely on Antarctic GIA inferences that have been derived with limited attention to full system and observational uncertainties (Ivins and James, 2005; Whitehouse et al., 2012b; Peltier et al., 2015). This limitation will propagate to the inferred magnitude of PD mass balance. These past reference GIA studies did not adequately consider the uncertainty relationship between the model and the actual physical system (model structural uncertainties) when quantifying data-model scores. At best, a limited exploration of parametric uncertainties on a very narrow set of ice sheet and GIA model ensemble parameters was performed with no assessment of model structural uncertainties. A data-constrained AIS and GIA model which accounts for complete uncertainties in the glacial system and Earth rheology is necessary to provide accurate bounds on contemporary mass balance of the AIS.

This study is the second part of a two-part study, and should be considered in conjunction with part one which analyzed history matched ice sheet evolution and fits to non-GIA data constraints. (Lecavalier and Tarasov, 2025). Part two of this study presented below aims to quantify bounds on the evolution of Antarctic GIA. This is carried out via an approximate history-matching methodology that explicitly accounts for data and model uncertainties. As part of the history-matching analysis presented in this 2 part study, a sub-ensemble of AIS simulations are chosen to represent the Antarctic component of the in progress GLAC3 (specifically version A denoted as GLAC3-A) global set of approximately history matched last glacial cycle ice sheet chronologies.

**2 Model description**

The GSM consists of comprehensive ice dynamic, climate forcing, and glacial isostatic components which are described in Tarasov et al. (2025) and Lecavalier and Tarasov (2025). To summarize the GSM includes: Hybrid SIA-SSA ice physics; with a subgrid grounding line ice flux parameterization; dual power basal drag for hard and soft bed sliding; ice shelf hydro-fracturing and restricted ice cliff failure; ocean temperature dependent sub ice shelf melt; subgrid ice shelf pinning point scheme; and a climate forcing representation with 14 ensemble parameters and fully coupled 2D energy balance climate model. An illustration showing the key components of the GSM is found in Figure S1 of Lecavalier and Tarasov (2025).

GIA models simulate the response of the solid Earth due to present and past changes in surface loading from the redistribution of ice, water, and mantle material. The two primary inputs to a GIA model are a global ice chronology and the Earth rheology. The GIA model products will henceforth be referred to as GIA inferences which include past and PD bedrock deformation, geoid and RSL estimates. The GSM is fully coupled to a GIA model of sea-level change based on a self-gravitating viscoelastic solid-Earth model which calculates GIA due to the redistribution of surface ice and ocean loads (Tarasov and Peltier, 1997) using a pseudo spectral solution for a spherically symmetric Earth rheology (Mitrovica and Peltier, 1991). The Earth model rheology has a standard Preliminary Reference Earth Model (PREM) density structure (Dziewonski and Anderson, 1981) which defines the radial elastic structure. The density structure is depth parameterized by volume averaging the values into shells with thickness of 10.5 km in the crust and 25 km in the mantle. Three ensemble parameters specify the lithospheric thickness and viscosity of the upper and lower mantles. The lithospheric thickness, upper mantle viscosity, and lower mantle viscosity can respectively vary between 46 to 146 km, $0.1 \cdot 10^{21}$ to $5 \cdot 10^{21}$ Pa·s, and $1 \cdot 10^{21}$ to $90 \cdot 10^{21}$ Pa·s. The GIA component shares many similarities to that used in Whitehouse et al. (2012b) for post-processing modelled ice sheet chronologies but in contrast to the GSM, their ice sheet model did not have this component coupled (but instead used a simple local relaxation response parametrization). Given typical GIA response timescales, the GIA calculations are computed every 100 simulation years. To minimize the considerable computational cost of solving for a complete gravitationally self- consistent solution coupled with an ice sheet model (Gomez et al., 2010, 2013), a zeroth order geoidal approximation is used to account for the gravitational deflection of the sea surface. This approximation sums all ice sheet contributions to the local geoidal deflection from the global mean as detailed in Tarasov et al. (2025). However, upon completing the simulation, a gravitationally self-consistent solution is computed using the AIS simulation in combination with interim GLAC3 chronologies for the other last glacial cycle ice sheets (e.g. Tarasov et al., 2012; Kageyama et al., 2017; Kierulf et al., 2021) as per the methodology of Mitrovica and Peltier (1991). The complete solutions are those that are compared against the GPS and RSL observations in Section 4. The full continental scale transient Antarctic simulations over 205 ka have a 40 by 40 km horizontal resolution with the full sea-level solution having a spherical harmonic degree and order of 512.

    As detailed in the accompanying study (Lecavalier and Tarasov, 2025), the Antarctic configuration of the GSM consists of 38 ensemble parameters. From the total 38 ensemble parameters, 10 are associated with ice dynamics (ice deformation, basal sliding), 11 with ice-ocean interactions (calving, sub-ice-shelf melt), 14 with ice-atmosphere interactions (atmospheric climate forcing), 3 with ice-solid Earth interactions (solid Earth rheology) and shown in Table 1 of Lecavalier and Tarasov (2025). Some of these ensemble parameters are applied to blend climate forcing schemes (parameterized PD climatologies, glacial index PMIP3 LGM climatologies, coupled energy balance climate model) to explore a wide range of plausible climate histories (Lecavalier and Tarasov, 2025; Tarasov et al., 2025). This represents the most comprehensive exploration of parametric uncertainties across the entire Antarctic glacial system of any study to date. A given simulation is defined by a parameter vector which consists of chosen values for each ensemble parameter. In this study, the GSM simulates AIS changes over the last 2 glacial cycles to minimize initialization uncertainties propagating into the last interglacial start of our history-matching interval. The GSM relies on several eustatic global sea-
level forcing time series (e.g. Lambeck et al., 2014; Lisiecki and Raymo, 2005) when performing
joint ice sheet and GIA calculations.

**3  Methodology**

History matching tests model simulations for inconsistency with observational constraints. This involves a data-model comparison that accounts for both data-system and system-model uncertainties (Tarasov and Goldstein, 2021). The observational constraint database applied in this research is the Antarctic ICe sheet Evolution observational constraint database version 2 (AntICE2 database; Lecavalier et al., 2023). The AntICE2 database is to date the largest quality-curated database of Antarctic paleo-data based on a variety of data types that can be leveraged to constrain various facets of the Antarctic glacial system. The database (Fig. 1) consists of observational data constraining past RSL, ice sheet thickness, ice sheet extent as well as PD observations of land motion from GPS measurements, ice core borehole temperature profiles, ice sheet geometry (Bedmachine version 2 Morlighem et al., 2020)  and ice surface velocity (Mouginot et al., 2019). The GSM simulations are scored against the highest quality data in the AntICE2 database, with a predominant focus on tier-1 and 2 data. Tier-1 data has the greatest power to constrain the ice sheet and GIA model and is deemed the highest quality data. Generally, tier-2 data provides more granular detail on past changes and supplements tier-1 data. Tier-3 data correlates highly with the higher quality tier-1/2 data, therefore, it is excluded from the history-matching analysis and only used for visual comparison. In this study, the data-model comparison focuses on the RSL and GPS data given its relevance to GIA processes. All other data-model comparisons and discussions can be found in the accompanying study (Lecavalier and Tarasov, 2025) which specifically focuses on the glaciological evolution of the AIS.

[Figure]

Figure 1: Antarctic continent and sector names mentioned in the study are shown alongside the Antarctic ICe sheet Evolution database version 2 (AntICE2) database (symbols). The data ID numbers for the paleo relative sea-level data (paleoRSL) and GPS data are shown. The remaining data ID information for the paleo ice extent (paleoExt), paleo ice thickness (paleoH), and borehole temperature data can be found in Figure 2 of Lecavalier et al. (2023). The Antarctic basemap was generated using Quantarctica (Matsuoka et al., 2021).

The paleoRSL and GPS data are inhomogeneously distributed across Antarctica in both space and time. The majority of the paleoRSL data spans the mid to late Holocene ages. The PD GPS measurements of bedrock displacement integrate the signal from several processes that operate on various time scales. The integrated time scale of the viscous relaxation due to ice and ocean unloading or loading depends on the viscosity of mantle material underlying the GPS station. Therefore, the paleoRSL and GPS bedrock displacement data most meaningfully constrain the Holocene Antarctic GIA. There are few places in Antarctica that are deglaciated and preserved RSL proxy data. Similarly, there are a limited number of GPS observations which have a significant signal-to-noise ratio with a minimal elastic correction due to contemporary mass loss.

Herein we only provide a cursory overview of history matching. For a comprehensive description we point the reader to Tarasov and Goldstein, 2021 (a future submission will explicitly detail the exact history-matching methodology). History matching yields a sub-ensemble of model simulations that are not-ruled-out-yet (NROY) by the data. The NROY sub-ensemble collectively provides bounds on the probable evolution of the AIS and GIA since the
last interglacial.

The history-matching methodology is diagrammatically illustrated in Figure S4 in Lecavalier and Tarasov (2025). It requires deep enough sampling of simulated chronologies to at least confidently "bracket reality". Each sample chronology is uniquely specified by a model ensemble parameter vector. The ensemble parameter prior ranges are based on numerical
experiments, previous studies, expert judgment, and are initially kept wide to improve capture of all potentially relevant regions of the parameter space.

When comparing a simulation to data, we carefully characterize the error model which combines all the errors attributed to data-system (measurement and indicative meaning uncertainties) and system-model (structural) uncertainties to produce a meaningful implausibility
score (Tarasov and Goldstein, 2021; Lecavalier and Tarasov, 2025). This includes a 5% structural bias error for RSL and a +1 mm/yr and -0.5 mm/yr bias error for PD vertical uplift both due to the use of a spherically symmetric global 1D Earth rheology instead of a 3D Earth rheology. These values were chosen on the basis of discrepancies between corresponding results for regional 1D and 3D Earth rheology modelling of Fennoscandia (Whitehouse et al., 2006;
Whitehouse, 2009). However, a similar analysis for Antarctica is lacking and needs to be addressed by the community.

As the robustness of history matching is predicated on the extent of sampling of the model parameter space, we use Markov Chain Monte Carlo (MCMC) to sample for the parameter vectors that are most likely to be consistent with the constraint data. MCMC sampling uses a
sequence of dependent autocorrelated samples generated through a Markov chain to approximate high dimensional probability distributions. Each sampling step in each chain requires a comparison of GSM predictions against the constraint data. To computationally enable the required multi-million-point MCMC sampling, GSM ensemble output is used for supervised training and validation of Bayesian Artificial Neural Networks (BANNs) to establish
computationally efficient emulators of the GSM. A BANN emulator is a probabilistic surrogate model with a specific architecture of interconnected layers of artificial neurons that approximates the behaviour of a complex system by learning from an ensemble of simulation outputs (Neal, 2012). The BANNs then efficiently provide estimates (along with associated uncertainties) for the GSM predictions as a function of the MCMC sample step ensemble parameter vector. The
BANN targets are key model metrics and/or specific predictions intended for data-model comparison. Many BANNs were trained to predict specific targets (e.g. grounded ice volume and area, past ice extent, ice thickness, RSL, GPS uplift rates) given parameter vectors and site coordinates as inputs. In the high-dimensional non-linear glacial system, it is challenging to ensure that all not implausible parameter space sectors have been identified. Consequently, this
study utilized hundreds of MCMC sampling chains that are initiated from widely dispersed points in the parameter space. A sub-sample of parameter vectors from the converged part of at least100 MCMC sampling chains is in turn used to create an ensemble of GSM simulations.

Convergence is assessed according to MCMC diagnostics as well from examining the evolution of a few of the state variables over the chain steps (Neal, 2012) . This combined MCMC and GSM ensemble iteration constitutes a "wave" of simulations that gets added to the full ensemble. The history-matching criteria rules out simulations with respect to the observations. As detailed in Tarasov and Goldstein (2021), this is achieved by comparing model output against observational constraints and ruling out simulations which are inconsistent with the constraint data after explicit accounting of relevant uncertainties. . Our implausibility threshold for inconsistency is a simulation-data misfit score component value of between 3-σ and 4-σ of the total uncertainty:

$$I = \frac{|d_i - E - \mu_{int} - \mu_{ext}|}{\sigma_{em} + \sigma_{struct} + \sigma_{obs}} \qquad [1]$$

The implausibility (I) includes the data-model residual ($d_i - E$) as well at the total internal and external discrepancy bias ($\mu_{int}, \mu_{ext}$) and all uncertainty sources: emulator ($\sigma_{em}$) structural ($\sigma_{struct}$), observational standard deviation ($\sigma_{obs}$) (see Table S1 in Lecavalier and Tarasov, 2025 for values, and Tarasov and Goldstein, 2021 for motivation). The implausibility threshold for each data type ($\sigma$ in eq.1) is applied to the corresponding data-model score component. In other words, a NROY simulation must be NROY for each data type. In the case of the Antarctic GSM configuration, the primary metrics of interest were chosen to be: PD ice thickness root-mean-square-error for West Antarctic Ice Sheet (WAIS; which includes the Antarctic Peninsula Ice Sheet for simplicity), East Antarctic Ice Sheet (EAIS), and floating ice; PD ice shelf area score; PD grounding-line position score along 5 transects; ice core borehole temperature profile score; GPS uplift rate score; past ice thickness score; past ice extent score; and past relative sea level score. To ensure an adequately sized NROY sub-ensemble, 3σ of the total uncertainty threshold was applied on all data type scores except for the following: past ice extent (3.5σ), floating ice RMSE (3.5σ), and relative sea-level scores (4σ). This gives an NROY set of 82 simulations (and corresponding parameter vectors). This larger allowance with these three scores was justified given uncertainties in the general properties of the underlying distribution (cf Tarasov & Goldstein, 2021) and given that the model struggles to bracket a few observations in this data class, which resulted in ruling out nearly all simulations if imposing a 3σ threshold across all data types.

Previous ensembles of simulations were evaluated against the AntICE2 database and PD observations to verify that the observations are adequately bracketed by the GSM given uncertainties (Lecavalier and Tarasov, 2025). To address any persistent data-model discrepancies following history-matching waves, the GSM underwent major model updates. A key refinement involved expanding the degrees of freedom in the ocean temperature forcing. Sub-ice-shelf mass balance in the GSM is computed using an ocean-temperature-dependent parameterization at the ice–ocean interface (Tarasov et al., 2025), encompassing the ice front, grounding line, and sub-shelf regions. Ocean temperature forcing is derived from transient TraCE-21ka simulations (He, 2011), bias-corrected using PD ECCO reanalysis data (Fukumori et al., 2018). For periods prior to 21 ka, a glacial index scheme is applied to the bias-corrected TraCE-21ka outputs. The temperature field beneath ice shelves is extrapolated with a depth cut-off based on minimum sill height to account for deeper continental shelves. To avoid extrapolating TraCE-21ka temperatures under warmer-than-present conditions, particularly relevant for Last Interglacial simulations, a separate ensemble parameter (rTOceanWrm) was introduced. This parameter scales glacial-index-derived atmospheric warming and adds it to the PD ocean climatology, leveraging the empirical relationship between Antarctic $\delta^2$H and mean ocean temperature (Shackleton et al., 2021). These enhancements, along with broader parameterizations of marine basins and other model components, led to an increase in the number of ensemble parameters, process additions to the GSM, and revisions to certain boundary conditions. Leading up to the final waves of ensembles, over 30,000 model simulations were performed as part of previous experimentation, sensitivity analyses, Latin Hypercube, and Beta fit sampling of ensemble parameters. In the results section, we present the latest iterations of large-ensemble results based on the history-matching analysis which consists of the final 9,293 simulations. This final waves of ensembles (N=9,293) is henceforth referred to as the "full ensemble" (see Table 2 in Lecavalier and Tarasov, 2025).

In addition to the history-matching analysis, an initial exploration on the potential impact of lateral Earth structure was conducted through a sensitivity analysis. In this study, reference to lateral Earth structure in the upper mantle is broadly defined wherever there is a gradient in upper mantle viscosity that exceeds approximately one order of magnitude on 100 to 1,000 km length scales (e.g. Figure 4 from Whitehouse et al., 2019). Past studies have found that the spatially averaged upper mantle viscosity to be on the order of $10^{20}$ to $10^{21}$ Pa·s (Ivins and James, 2005; Whitehouse et al., 2012b) which is within the range evaluated as part of the history-matching analysis. However, there are more recent estimates of an anomalously low upper mantle viscosity for the Antarctic Peninsula, Amundsen sector, and part of the Weddell and Ross Sea sector on the order of $10^{18}$ to $10^{19}$ Pa·s (Wolstencroft et al., 2015; Zhao et al., 2017; Barletta et al., 2018; Nield et al., 2018; Whitehouse et al., 2019). A high variance 18 member subset (HVSS) of simulations were selected from the NROY sub-ensemble according to key metrics of interest, such as the AIS grounded ice volume during the Last Glacial Maximum (LGM) (Figure S7 in Lecavalier and Tarasov, 2025). Each metric of interest (last interglacial deficit and LGM excess relative to present) and AntICE2 data type scores were respectively normalized, and a simulation was chosen from the NROY subensemble to initialize the HVSS sampling (e.g. a NROY simulation with minimum of the maximum score across all primary data types). Each subsequent sample added to the HVSS is selected by identifying which simulation in the NROY subensemble maximizes the multidimensional distance (square root sum of squares) between all the normalized metrics and scores against the simulations already populating the HVSS. This method extracts a subset of simulations which exhibit a wide range of behaviours across the NROY sub-ensemble. This also included selecting simulations with minimum scores to certain data types. The ice load chronologies from the 18 members of the NROY sub-ensemble HVSS were subject to repeated GIA post-processing over a range of Earth models. This involved applying each individual ice load chronology in the HVSS as input to the gravitationally self-consistent sea level solver using alternate Earth models. Specifically, the lithospheric thickness and upper mantle viscosity was progressively decreased from 146 km and $5 \cdot 10^{21}$ Pa·s to 46 km and $5 \cdot 10^{18}$ Pa·s (146, 120, 96, 71, to 46 km; $5 \cdot 10^{21}$, $5 \cdot 10^{20}$, $5 \cdot 10^{19}$, to $5 \cdot 10^{18}$ Pa·s), respectively, to evaluate the impact of an anomalously low upper mantle viscosity on isostasy (Fig. S1 and S2). This experimental design isolates the Earth model sensitivity at the cost of lost dynamical self-consistency between the ice history and Earth model.

[Figure]

Figure 2: Paleo relative sea level data-model comparison where the grey shading are the full ensemble statistics. The solid and dashed black lines are the mean and min/max ranges for the not-ruled-out-yet (NROY) best fitting sub-ensemble. Simulations consisting of a high variance subset (HVSS) of the NROY sub-ensemble are shown in red. The 2σ and 1σ ranges are the nominal 95% and 68% ensemble intervals based on the equivalent Gaussian quantiles, respectively.

**4 Results and Discussion**

Below we present the full ensemble and NROY sub-ensemble against the AntICE2 observational constraints of most relevance to GIA: past RSL and elastic-corrected GPS measured rates of vertical land motion. With regards to the latter, the GPS measurements have to be corrected for the elastic response due to recent ice mass changes to isolate the viscous signal due to past load changes (Martín-Español et al., 2016; Sasgen et al., 2017). A challenge is the validity of the elastic corrections which are dependent on the inferred contemporary ice load changes which have their own explicit and some ill-defined implicit uncertainties. To mitigate this issue, only GPS data that is minimally impacted by a contemporary elastic signal are considered. 
[revised manuscript text omitted]
. This could be attributed to increase precipitation or ice being advected to the region which thickens the ice column in the late Holocene. Conversely, 8502 with its exceedingly high elastic-corrected uplift rate indicates significant mass loss in the late Holocene. A large quantity of regional ice mass loss can be linked to ocean forcing and margin retreat of marine-based ice overlying soft till during the late Holocene. This suggests that
the GSM might have insufficient degrees of freedom in the regional climate forcing and basal environment to produce a sufficiently late ice load scenario to reconcile these remaining discrepancies.

**4.2 Glacial isostatic adjustment model predictions**

The history-matching result being the NROY sub-ensemble is a product of ruling out simulations that were inconsistent (within 3 to 4σ) with all the data types in the AntICE2 database. A given data type in the database constrains the Antarctic GIA ensemble results to various degrees since some data types are direct constraints on GIA (e.g. bedrock displacement) while others are indirect (e.g. load history). The NROY sub-ensemble brackets the majority of
the AntICE2 database with some limited outstanding exceptions discussed above and in Lecavalier and Tarasov (2025). The full ensemble is sieved on a data type basis which avoids the need for any inter data type weighing. History matching on its own does not produce a probability distribution of chronologies. Throughout this study, we present the NROY sub-ensemble nominal 2σ range since studies that apply GIA corrections are typically interested in
accounting for nominal 2σ uncertainties. Moreover, by visualizing 2σ ranges one avoids any one outlier simulation from dominating the visualization. However, these nominal ranges should not be confused with traditional Gaussian confidence intervals and the reader is encouraged to also consider the complete NROY sub-ensemble min and max GIA and RSL ranges shown in Figure S3-S6. When evaluating the spatial RSL, uplift rates, and geoid NROY sub-ensemble plots
(Figure 4, 5, 6, and 7), the min/max and -2σ/2σ plots differ primary by magnitude where the NROY sub-ensemble min is broadly more negative than those of the -2σ fields (conversely true for the NROY sub-ensemble max and 2σ fields). The difference between the min/max and -2σ/2σ plots illustrate the impact of edge case simulations that made it within the NROY sub-ensemble which can be considerable in certain regions (e.g. >10 m difference in WAIS RSL at
10 ka between the max and 2σ field as shown in Figure 4f and S3f).

[revised manuscript text omitted]
. Additionally, the sensitivity analysis does not consider coupled GIA feedbacks with these anomalously low upper mantle viscosities. Even though the NROY sub-ensemble is consistent with the entire AntICE2 database and represents a wide range of ice loading chronologies, 3D Earth GIA models could produce responses to ice loading that are not bracketed by the NROY sub-ensemble due to GIA feedbacks caused by lateral Earth structure.

Thus, the NROY sub-ensemble GIA predictions likely underestimate the uplift and geoid rate bounds in specific regions with anomalous viscosity structure. A future history-matching analysis should apply a broader range of upper mantle viscosity Earth models to include values down to $10^{19}$ pa·s (van der Wal et al., 2015) as applied in our sensitivity analysis given that WAIS rests atop several anomalously low viscosity zones and this region experienced the greatest mass change since the last interglacial. However, it remains unclear to which degree it is appropriate to use such low viscosity values for glacial cycle simulations given that model-based support for use of such a low viscosity in visco-elastic models coupled to ice sheet models without lateral Earth viscosity variation has only been shown for contemporary load change contexts (van der Wal et al., 2015).

[Figure]

Figure 6: Present-day Antarctic rate of geoid displacement for the not-ruled-out-yet (NROY) sub-ensemble a) mean, b-c) minus and plus 2σ. The geoid trend presented above only includes the Antarctic contribution to geoid perturbations and does not include global eustatic contributions. Three glaciology self-consistent simulations chosen from a NROY high variance subset (HVSS): d) RefSim1 (NROY member with minimum score across all the data types in AntICE2); e) RefSim4 (NROY member with minimum GPS score); f) RefSim5 (NROY member with minimum score across all the paleo data types in AntICE2). The 2σ ranges are the nominal 95% ensemble intervals based on the equivalent Gaussian quantiles. The RefSims PD simulated grounding line is shown by the black contour.

[Figure]

Figure 7: The (a) minimum and (b) maximum bounds for the PD rate of bedrock displacement for the three reference Antarctic GIA inferences (IJ05_R2 - Ivins and James, 2005; W12a - Whitehouse et al., 2012b; ICE-6G_D - Peltier et al., 2015). These three GIA inferences represent nominal 2σ bounds on the PD GIA corrections applied in the IMBIE studies to infer contemporary mass balance of the AIS. (Shepherd et al., 2018; Otosaka et al., 2023). The not-ruled-out-yet (NROY) sub-ensemble 2σ (c) lower and (d) upper bounds minus the respective bounds of the three reference GIA inferences. The differences shown in (c) and (d) demonstrate regions where the three reference GIA inferences underestimate PD GIA uncertainties or where the NROY sub-ensemble better constrain the regional GIA response relative to the three reference GIA inferences.

**5 Conclusion**

In this study a sub-ensemble of Antarctic GIA inferences is presented based on a history-matching analysis of the GSM against the AntICE2 database. The fully coupled glaciological and GIA model was used to generate a full ensemble consisting of 9,293 Antarctic simulations spanning the last 2 glacial cycles. BANNs were trained to emulate the GSM for rapid exploration of the parameter space via MCMC sampling. Simulation results were scored against past relative sea level, PD vertical land motion, past ice extent, past ice thickness, borehole temperature profiles, PD geometry and surface velocity. The full ensemble of simulations broadly brackets the AntICE2 database with a few outstanding data-model discrepancies likely attributed to model resolution, insufficiently climate forcing degrees of freedom for certain sectors, and insufficient accounting for uncertainties in the basal environment. In particular, this manifest in a few outstanding data-model discrepancies in regions with inadequately resolved complex topography such as the Transantarctic Mountains or regions with likely many subgrid pinning points that can help stabilize an ice shelf and grounding line. The scores were used in the history-matching analysis to rule out simulations that were inconsistent with the data given observational and structural uncertainties, thereby a NROY sub-ensemble (N=82) that bounds past and present GIA and sea-level change was generated.

Given that our history matching accounts for data-system and system-model uncertainties to a much deeper extent than any previous AIS study, the NROY sub-ensemble provides the most credible bounds to date on actual Antarctic GIA and last glacial cycle ice sheet evolution. As such, our analysis demonstrates that previous Antarctic GIA studies have underestimated the viscous deformation contribution to PD uplift rates due to past ice sheet changes across several key regions. This is particularly the case in the Amundsen sector, an area currently undergoing significant mass loss, which has a large range of viable PD GIA estimates. Our NROY set of chronologies will therefore facilitate more accurate inference of the PD mass balance of the AIS, including for vulnerable marine-based regions.

The NROY sub-ensemble of AIS results represent a collection of not-ruled-out-yet Antarctic components for the in progress global GLAC3 set of last glacial cycle ice sheet chronologies. Future research will prioritize a history-matching analysis using a higher horizontal resolution Antarctic configuration of the GSM, the integration of additional observational constraints such as the age structure of the ice inferred from reflective isochrones in radiostratigraphic data, and a 3D Earth viscoisty GIA emulator (Love et al., 2024) to better represent lateral Earth structure.

**Author Contribution**

[revised manuscript text omitted]

McKay, D. I. A., Staal, A., Abrams, J. F., Winkelmann, R., Sakschewski, B., Loriani, S., Fetzer, 830 I., Cornell, S. E., Rockström, J., and Lenton, T. M.: Exceeding 1.5°C global warming could trigger multiple climate tipping points, Science (1979), 377, https://doi.org/10.1126/science.abn7950, 2022.

Mitrovica, J. X., and Peltier, W. R.: A complete formalism for the inversion of post-glacial rebound data: resolving power analysis. Geophysical Journal International, 104(2), https://doi.org/10.1111/j.1365-246X.1991.tb02511.x, 1991.

[revised manuscript text omitted]

van der Wal, W., Whitehouse, P.L. and Schrama, E.J.: Effect of GIA models with 3D composite mantle viscosity on GRACE mass balance estimates for Antarctica. Earth and Planetary Science Letters, 414, pp.134-143, 2015.

Velicogna, I., Mohajerani, Y., Geruo, A., Landerer, F., Mouginot, J., Noel, B., Rignot, E., Sutterley, T., van den Broeke, M., van Wessem, M., and Wiese, D.: Continuity of Ice Sheet Mass Loss in Greenland and Antarctica From the GRACE and GRACE Follow-On Missions, Geophys Res Lett, 47, https://doi.org/10.1029/2020GL087291, 2020.

Whitehouse, P., Latychev, K., Milne, G. A., Mitrovica, J. X., and Kendall, R.: Impact of 3-D Earth structure on Fennoscandian glacial isostatic adjustment: Implications for space-geodetic estimates of present-day crustal deformations. Geophysical Research Letters, 33(13), 2006.

Whitehouse, P.: Glacial isostatic adjustment and sea-level change, State of the art report, 2009.

Whitehouse, P. L., Bentley, M. J., and Le Brocq, A. M.: A deglacial model for Antarctica: geological constraints and glaciological modelling as a basis for a new model of Antarctic glacial isostatic adjustment. Quaternary Science Reviews, 32, 1-24. https://doi.org/10.1016/j.quascirev.2011.11.016, 2012a.

Whitehouse, P. L., Bentley, M. J., Milne, G. A., King, M. A., and Thomas, I. D.: A new glacial isostatic adjustment model for Antarctica: Calibrated and tested using observations of relative sea-level change and present-day uplift rates, Geophys J Int, 190, 1464–1482, https://doi.org/10.1111/j.1365-246X.2012.05557.x, 2012b.

Whitehouse, P. L.: Glacial isostatic adjustment modelling: Historical perspectives, recent advances, and future directions, Earth Surface Dynamics, 6, 401–429, https://doi.org/10.5194/esurf-6-401-2018, 2018.

Whitehouse, P. L., Gomez, N., King, M. A., and Wiens, D. A.: Solid Earth change and the evolution of the Antarctic Ice Sheet, https://doi.org/10.1038/s41467-018-08068-y, 1 December 2019.

Wolstencroft, M., King, M.A., Whitehouse, P.L., Bentley, M.J., Nield, G.A., King, E.C., McMillan, M., Shepherd, A., Barletta, V., Bordoni, A. and Riva, R.E.: Uplift rates from a new high-density GPS network in Palmer Land indicate significant late Holocene ice loss in the southwestern Weddell Sea, Geophysical Journal International, 203(1), 737-754, 2015.

Zhao, C., King, M.A., Watson, C.S., Barletta, V.R., Bordoni, A., Dell, M., and Whitehouse, P.L.: Rapid ice unloading in the Fleming Glacier region, southern Antarctic Peninsula, and its effect on bedrock uplift rates, Earth and Planetary Science Letters, 473, 164-176, 2017.